# An Updated Comprehensive Overview of Different Food Applications of W_1_/O/W_2_ and O_1_/W/O_2_ Double Emulsions

**DOI:** 10.3390/foods13030485

**Published:** 2024-02-02

**Authors:** Fatemeh Ghiasi, Hadi Hashemi, Sara Esteghlal, Seyed Mohammad Hashem Hosseini

**Affiliations:** Department of Food Science and Technology, School of Agriculture, Shiraz University, Shiraz 71441-13131, Iran; hadihashemigahruie@shirazu.ac.ir (H.H.); esteghlalsara@gmail.com (S.E.); hhosseini@shirazu.ac.ir (S.M.H.H.)

**Keywords:** double emulsion, food application, encapsulation, fat reduction, emulsified edible film

## Abstract

Double emulsions (DEs) present promising applications as alternatives to conventional emulsions in the pharmaceutical, cosmetic, and food industries. However, most review articles have focused on the formulation, preparation approaches, physical stability, and release profile of encapsulants based on DEs, particularly water-in-oil-in-water (W_1_/O/W_2_), with less attention paid to specific food applications. Therefore, this review offers updated detailed research advances in potential food applications of both W_1_/O/W_2_ and oil-in-water-in-oil (O_1_/W/O_2_) DEs over the past decade. To this end, various food-relevant applications of DEs in the fortification; preservation (antioxidant and antimicrobial targets); encapsulation of enzymes; delivery and protection of probiotics; color stability; the masking of unpleasant tastes and odors; the development of healthy foods with low levels of fat, sugar, and salt; and design of novel edible packaging are discussed and their functional properties and release characteristics during storage and digestion are highlighted.

## 1. Introduction

Double emulsions (DEs), also defined as an emulsion-in-emulsion structure, are a major type of multiple emulsions (MEs) containing complex multiphase colloidal structures in which small internal water droplets are dispersed within larger oil droplets that are then entrapped in an external continuous water phase [1,2]. According to Scopus results, the ratio of publications on DEs and MEs increased significantly, by 20.18%, during the last decade (Figure 1), mainly due to their superior characteristics in co-encapsulation of both hydrophilic and lipophilic food ingredients and low-fat diet design. Generally, double emulsions are classified into two main categories, water-in-oil-in-water emulsions denoted by W_1_/O/W_2_ (Figure 2a) and oil-in-water-in-oil emulsions denoted by O_1_/W/O_2_ (Figure 2b), to distinguish between two aqueous and oil phases with different compositions [3]. However, various types of Des, such as W/W/O, W/O/O, O/W/W, and O/O/W, have also been discovered through the use of advanced technologies [4].

Various emulsification methods for the preparation of DEs, including high-speed shearing, mainly using rotor-stator devices; high-pressure homogenization; membrane emulsification; ultrasonication; and microfluidics, are comprehensively reviewed in the literature [3]. The simultaneous application of both hydrophilic and lipophilic food-grade emulsifiers, with high and low hydrophilic–lipophilic balance (HLB) values, respectively, is the most critical condition for the successful preparation of conventional multiple emulsions [1]. However, stabilization mechanisms that use dispersed colloidal particles, called Pickering emulsions, as well as sol-gel approaches, have recently garnered increased interest [5,6].

Among DEs, the W_1_/O/W_2_ type of DEs have more promising applications in food-related fields due to the possibility of their replacement with conventional oil-in-water (O/W) emulsions to formulate various reduced-fat products while maintaining a favorable mouthfeel and texture [7]. Additionally, the potential of W_1_/O/W_2_ DEs for fortification purposes via encapsulation, targeted delivery, controlled-release, and enhanced bioavailability of various sensitive hydrophilic essential nutrients (e.g., vitamins, polyphenols, extracts, flavors, minerals, coloring agents, etc.) during food storage as well as consumption and digestion have been discussed in recent research [8]. It has been reported that the entrapment of hydrophilic compounds within W_1,_ as compared to the external aqueous phase (W_2_), may have positive consequences such as protection against chemical degradation during processing (e.g., the heat-induced degradation of vitamins) and storage, preventing potential reactions with other water-(in)soluble components (e.g., the separation of some phenolic compounds from proteins or the separation of transition metals from lipophilic ingredients), and masking undesirable sensory attributes (e.g., astringency, bitterness, or a metallic aftertaste) during mastication, hence increasing the acceptability of foods [9,10]. The enhancement of the perception of saltiness and sweetness in food formulations using salt and sucrose encapsulation within the internal aqueous phase of Des, as well as the masking unpleasant taste characteristics of some water-soluble components, have also been reported on as promising W_1_/O/W_2_ applications [11,12].

There are several reviews on the preparation, stability, and characterization of multiple emulsions, most of which are restricted to W_1_/O/W_2_. To our knowledge, this review is the first attempt to produce an updated comprehensive overview of the preceding decade’s publications on the applications of both W_1_/O/W_2_ and O_1_/W/O_2_ DEs across different aspects of the food industry, including fortification purposes with nutraceuticals, probiotic and enzyme encapsulation, food preservation in terms of improved antioxidant and antimicrobial properties, food packaging, the development of healthy reduced-fat/sugar/salt foods, and the improvement of sensory properties in terms of color stability and the masking of unpleasant flavors.

## 2. Food Applications of W_1_/O/W_2_ DEs

### 2.1. Fortification Purposes

The formulation of nutritionally fortified foods has gained considerable importance due to the increasing demand for health beneficial foods. However, some food ingredients and micronutrients with health promoting properties, such as vitamins, minerals, and polyphenols, are sensitive to environmental conditions (e.g., pH, light, oxygen, heat, etc.) and, hence, cannot be added directly to foods. DE is an appropriate carrier for micronutrients that, in addition to protecting them against degradation, provides the possibility of co-encapsulation of both hydrophilic and hydrophobic compounds to produce enhanced health benefits [13], as evidenced in Table 1. In this context, Dai et al. [14] co-encapsulated vitamin C and β-carotene within the W_1_ and oil phase of a DE, respectively; stabilized by different concentrations of *Sipunculus nudus* salt-soluble proteins. The values of encapsulation efficiency (EE) for vitamin C at 1% and 2% of protein concentrations were 87.3% and 91.2%, respectively. While β-carotene showed more EE (99.7% and 99.8%), which was not affected by protein concentration. Measuring the antioxidant ability of the samples after 28 days of storage at 4, 37, and 55 °C showed a decrease in the antioxidant ability of vitamin C and β-carotene during this time, especially at higher temperatures. However, comparing the encapsulated and free vitamin C and β-carotene indicated the excellent ability of the DE to protect them against degradation, resulting in higher antioxidant ability.

Co-encapsulation of hydrophobic astaxanthin and hydrophilic phycocyanin in a pH-responsive DE-filled gellan was carried out by Yu, et al. [13]. The highest EEs of astaxanthin (90.82%) and phycocyanin (94.1%) were achieved at 0.5% and 0.7% gellan concentrations, respectively. Examining the storage stability of the free and encapsulated active compounds showed attenuation rates of 79.9% and 48.1% for solutions of astaxanthin and phycocyanin, respectively, after 10 days of storage; while 25.6% and 16.8% degradation was reported in the encapsulated forms after the same time. In vitro release digestion showed that the DE structure remained intact in acidic conditions with a maximum 25% release of the total encapsulated astaxanthin and phycocyanin. In simulated intestinal fluid (SIF), due to the pH-responsive activity of gellan, the DE structure was destroyed under the effect of an alkaline pH, resulting in higher releases of astaxanthin and phycocyanin (>60%). In another work, Barbosa and Garcia-Rojas [15] encapsulated iron in the inner phase of a DE and investigated its release and bioaccessibility in adults and infants. The DE samples were prepared using different concentrations of whey protein isolate (WPI), polyglycerol polyricinoleate (PGPR), tara gum, and sucrose. Increasing WPI led to a higher EE of iron. The addition of sucrose caused a higher EE and, therefore, the highest EE (96.9%) was exhibited by the sample with 12% WPI and 2% sucrose after preparation. The EE decreased for all samples after 7 days of storage; however, the lowest reduction was obtained at 2% sucrose. In vitro digestion in adults occurred in the oral (8.36% release), gastric (38.56% release), and intestinal (51.47% release) stages, while in infants, in vitro digestion and release of iron occurred under gastric (27.22%) and intestinal (41.45%) conditions. In addition, the bioaccessibility of iron after digestion was 49.54% for adults and 39.71% for infants. Encapsulation of Magnesium (Mg) in the inner phase of the DE was also performed to make a fortified Mg cake by Kabakci et al. [16]. Lentil flour was used as a stabilizing agent in the DE. It was observed that increasing the lentil flour decreased EE, so that at 15% and 30% lentil flour, the EE was 97.54% and 92.42%, respectively. Using a high-shear homogenizing method of emulsification resulted in higher EE than ultrasonic homogenization. The addition of Mg to the DE and single emulsion (SE) led to lower hardness than free Mg. The volume of the cake was not affected by the DE and the weight losses of the cakes with DE and SE were higher than of that containing free Mg. The crust color of the three cakes was similar. Sensory analysis showed the lowest score for the cake with free Mg and the taste of the cakes with DE was similar to those without Mg, indicating the ability of DE to mask the unpleasant taste of Mg. The heat protection of the DE was 12.5% higher than the SE during the baking process and 79–83% of the Mg was preserved by the DE at different lentil flour concentrations during baking. In vitro digestion of the Mg in the DE and free Mg samples was similar and both the encapsulated and free Mg were digested under gastric and intestinal conditions. Su et al. [17] encapsulated different amino acids in a DE stabilized by gum Arabic (GA) and xanthan. It was observed that EE was affected by xanthan content and, at concentrations higher than 0.3%, the EE decreased. Therefore, a DE with 50% W_1_/O, 2% gum Arabic, and 0.3% xanthan was prepared. The EE values of aspartic acid, glycine, and lysine as acidic, neutral, and basic amino acids were 82%, 92%, and 83%, respectively. The highest release rate was reported for glycine at 25 °C after 28 days of storage. After 14 days, more than 50% of the glycine was released. The lowest content release rate was reported for lysine (<20% after 28 days). Amino acid leakage was lower at 4 °C compared to 25 °C, resulting in the release of only 40% of glycine with more hydrophobicity after 28 days. Li et al. [18] used a W_1_/O/W_1_ DE for vitamin B_12_ encapsulation in the W_1_. The EE and in vitro digestion were assessed under the effects of the W_1_/O-to-W_2_ ratio and stabilizers of the outer emulsions (Tween 20, soy lipophilic protein (LP), and a LP-methyl cellulose (MC) complex). Using LP and LP-MC resulted in higher vitamin B_12_ EE than Tween 20. The reduction in EE after 14 days of storage was higher in samples containing Tween 20. An LP: MC ratio of 3:1 favored EE and resulted in a higher value. During in vitro digestion, continuous release of vitamin B_12_ occurred. At an LP: MC ratio of 1:3, 60% of the vitamin B_12_ leaked (oral stage) and the highest release was recorded for the Tween 20-stabilized DE in this stage. Meanwhile, the sample with a W_1_/O-to-W_2_ ratio of 4:6 and an LP: MC ratio of 3:1 showed sustained release during digestion.

Hosseini et al. [19] investigated the effects of different heat treatments, including ohmic, microwave, and conventional methods, on an iron-loaded DE. Demulsification and visible phase separation were reported to be affected by microwave treatment. Ohmic treatment caused more iron release and lipid oxidation than conventional heating. Regarding the iron bioavailability in cell line SW742, heat treatment increased the bioavailability from 37.28% to 42.94%. As expected, yogurt samples fortified with iron-loaded DEs presented higher lipid oxidation during storage than those incorporating an iron-free DE. Dima and Dima [20] co-encapsulated vitamin D_3_ and calcium salts (e.g., carbonate, tricalcium phosphate, and calcium gluconate with various solubilities) into a DE and evaluated their bioavailability in vitro under simulated gastrointestinal conditions. To this end, three designs of DEs were developed: gelled water in an oil-in-water (W_1G_/O/W_2_) DE using sodium alginate (SA) in W_1_, W_1_/O/W_G2_ DEs with individual alginate and chitosan in W_2_, and a W_1_/O/W_2_ DE encapsulated in alginate/chitosan microparticles. The particle size of all DEs after digestion in the oral phase increased due to the coalescence and flocculation phenomena, while it decreased significantly in the gastric and intestinal phases. The W_1_/O/W_G2_ DEs stabilized with alginate showed lower calcium release than those stabilized with chitosan due to its ionized carboxyl groups. Generally, lower rates of calcium release were reported in the W_1_/O/W_2_ DE loaded in microparticles than free W_1_/O/W_2_ DEs, due to erosion. Moreover, the presence of calcium ions in the intestinal fluid decreased both the free fatty acid content and the bioaccessibility of vitamin D_3_ due to the preventative micellization process. Vladisavljević et al. [21] developed copper-loaded food-grade DEs with a high entrapment efficiency (95−97%) and narrow particle size distribution (span = 0.36−0.56) using an oscillating membrane emulsification technique. They reported an initial burst release due to the composition ripening followed by a sustained release during storage as a result of molecular diffusion through the oil phase. Moreover, a larger particle size resulted in a lower release rate of copper followed by non-zero-order kinetics.

In conclusion, the high sensitivity of bioactive compounds such as vitamins, minerals, amino acids, etc., as well as their possible adverse interactions with other food ingredients, are considered to be one of the main concerns in the food industry. The above publications confirmed that DEs can be used for efficient (co-)encapsulation of different sensitive compounds to protect them against degradation and increase their bioavailability in fortified food products without significant effects on the food’s sensory properties. The release of these bioactives can be modulated by using appropriate biopolymers in DE formulation to design target release systems. Therefore, fortification purposes have been the most common applications of DEs in recent decades.

**Table 1 foods-13-00485-t001:** Overview of the past decade’s publications on the fortification applications of W_1_/O/W_2_ double emulsions (DEs).

W_1_	O	W_2_	Phase Ratio		Remarkable Results	Reference
			W_1_:O	W_1_/O:W_2_		
- **Vitamin B_12_** (0.2%) - NaCl (0.1 M) - Different gelators (1%) including xanthan gum, gum Arabic, methylcellulose, and pectin	- Olive oil - PGPR (8%)	- Soybean lipophilic protein LP (4%) - Different gelators (2%) including xanthan gum, gum Arabic, methylcellulose, and pectin	25:75	30:70	- Droplet size during in vitro digestion: 0.67–1.91 µm (after oral), 0.88–2.06 (after stomach), 0.42–0.52 (after intestines) - **↑** The EE after W_1_ gelation (>90%) and internal aqueous-phase retention (more than 85%) - ↓ Interfacial tension after W_1_ and W_2_ gelation - ↑ Vitamin B_12_ antioxidant activity, particularly after W_2_ gelation - ↑ Digestion performance of emulsions, emulsion stability, sustained release profile, and bioavailability after W_1_ and W_2_ gelation, particularly after W_2_ gelation	[22]
- **Sesamol (0.04%)** - Phosphate-buffered saline	- Olive oil - **Retinol (0.003%)** - Span 80 (5–15%)	- Tween 80 (1–10%) - Alginate (0.067%)	20:80 30:70 40:60	Not mentioned	- Optimized formulation with a total EE of 92.93% and particle size of 381.94 nm: 6.24% Tween 80, 10.84%, 37.70% Span 80, and a 40:60 ratio of W_1_/O - Results of chitosan-coating of optimum sample: - DSC: endothermic peak at 236.48 °C due to the ionic interactions of Alginate-chitosan - In vitro release of sesamol and retinol: 39% and 22% in simulated gastric fluid (SGF), respectively, and 56% and 22% in simulated intestinal fluid (SIF), respectively.	[23]
- Flaxseed gum (0.6%) - NaCl (100 mM) - **Collagen peptide** (10%)	- Soybean oil - Rice bran wax (0–7%) - PGPR (2%) - **Astaxanthin** (0.1%)	- Flaxseed gum/Whey protein isolate (WPI) complexes (1%) - NaCl (100 mM)	40:60	10:90 20:80 30:70 40:60 50:50	- ↑ Storage stability at 25 °C for 28 days - ↑ Particle size from 37.38 μm to 94.98 μm by ↑ rice bran was from 1% to 7% - Results of DE-filled hydrogel beads: - ↑ gel strength, EE, and bioaccessibility of collagen peptide and astaxanthin by the addition of higher rice bran wax - The highest bioaccessibility of astaxanthin (93.80%) at 30% W/O - Slow release in SGI while a rapid release in SIF - ↑ Rice bran wax led to ↓ release rate of bioactives in digestion stage	[24]
- NaCl (0.9%) - **Low molecular weight oyster peptides** (LOPs) (40%)	- Mixture of sunflower oil, MCT, Fish oil (2:2:1) - PGPR (8%)	- Mixture of WPI, maltodextrin, and fructooligosaccharides (2:2:1) - Tween 80 (0.25%)	40:60	15:85	- The highest stability, smallest particle size (≈2 µm), highest whiteness (77.99) and turbidity (30,773), and lowest viscosity at 125 W sonication power - The highest emulsifying activity index (1.61) and emulsifying stability index (98.37) at 125 W - ↑ Conjugated dienes during storage time up to 20 days followed by a ↓ trend to 30 days, low levels at sonication power > 100 W - Well-masked fishy odor at 125 W	[25]
- **Ferric sodium EDTA or ferrous sulphate heptahydrate** (3%)	- Corn oil - PGPR 90 (5%) - **Curcumin** (0.1%)	- Different biopolymers including sodium caseinate (SC) (2%)/sodium alginate (SA) (1%), SC (2%)/sodium carboxymethyl cellulose(NaCMC) (1%), and SC (2%)/β-cyclodextrin (βCD) (1%)	40:60	22:100	- Particle size: 4.6, 5.1, and 8.0 µm for SC/NaCMC, SC/SA, and SC/βCD, respectively - The highest EE of ferric sodium EDTA (86.5%), ferrous sulphate heptahydrate (94%), and curcumin (81.4%) in the presence of SC/βCD, SC/NaCMC, and SC/SA, respectively - ↑ Bioaccessibility of ferric iron(~86–92%) and curcumin (~76–86%) - ↑ The low intestinal bioaccessibility of curcumin (~33%) to 69.5% for SC/βCD - The highest bioaccessibility of ferric sodium EDTA for complexes of SC/SA and SC/NaCMC, i.e., 92.2 and 89.7%, respectively. - High bioaccessibility of both ferric iron (~86–92%) and curcumin (~76–86%) for SC/NaCMC and SC/SA	[26]
- **Vitamin B_12_** (0.1%)	- Sunflower oil - PGPR (2–10%)	- Gum Arabic (0.5–8%)	30:70	10:90	- ↓ Particle size and surface tension by ↑ PGPR - ↓ Zeta potential and surface tension by ↑ gum Arabic, while particle size first increased followed by a decreasing trend at more than 20% - ↑ Release rate of FFA by ↑ gum Arabic - EE and release efficiency of vitamin: 80% and 95% at 8% gum, respectively.	[27]
- Gelatin (1%) - **Vitamin C** (0.05–1%)	- Camellia oleifera seed oil - PGPR (5%)	- Phosphate Buffer - Gelatin-EGCG-high methoxyl pectin complex (10 mM)	20:80	60:40	- Droplet size: 0.5–1.55 µm, ↑ at higher vitamin content - Zeta: −33.73 to −38 mV, ↓ after vitamin addition - ↓ EE with ↑ vitamin, the highest EE (90.23%) at 0.05% - ↑ DE stability at alkaline pH and ↓ stability after sodium ion addition - ↑ DEs storage stability after vitamin addition for 15 days - Sustained-release function of vitamin C, slow release in SIF - Bioavailability after simulated digestion: 25% which was higher than that in W_1_/O - Fatty acid release: W_1_/O > DE	[28]
- Different amino acids including **aspartic acid, glycine, and lysine** (2.5%)	- Corn oil - PGPR (6%)	- Gum Arabic (0.5–6%) - Xanthan gum (0–2%)	50:50	20:80 50:50	- Particle size: 154 µm and 42–33 µm without and with xanthan - Rheology: *G′* > *G″* for all samples - The highest viscosity coefficient (135.21 Pa.s^n^) at 2% and the lowest at 0% (8.53 Pa.s^n^) xanthan - The highest EE (82–92%) at 50% W_1_/O, 2% gum Arabic and 0.3% xanthan - Higher release of hydrophobic amino acids than hydrophilic ones, higher release rate at 25 °C than 4 °C	[17]
- **Insulin** (20%)	- Soybean oil - **Quercetin** (0.1%) - PGPR (5%)	- Black soybean protein-SA (BBP-SA) conjugate (5%)	30:70	30:70	- Formation of emulsion gels by adding CaCl_2_ (0–0.04 M), gluconolactone (GDL, 0–2%), and glutamine transaminase (TGase, 0–20 U/g) - ↑ WHC by increasing gelling agents but ↓ > 0.03 M CaCl_2_ Droplet size: 1–10 µm - ↑ EE for insulin and quercetin until they reached a maximum (96.74% and 91.35%, respectively) at a 0.03 CaCl_2_. - ↑ EE for both insulin and quercetin until they reached a maximum (97.06% and 95.82%, respectively) at 20% TGase - ↑ EE for both insulin and quercetin at higher GDL	[29]
- **Insulin** (0.1 M)	- Soybean oil - Quercetin (1 mg/L) - PGPR (5%)	- Different emulsifiers including black bean protein (BBP), Tween 80, lecithin, and pectin (2%)	30:70	30:70	- Order of particle size: lectin > pectin > BBP > Tween 80 - The highest zeta potential value (52.80 mV) and EE (insulin: 95.7%, quercetin: 93.4%), emulsifying properties, temperature stability, lower levels of PV (0.86 mM), and TBARS (25.80 μM) for BBP-stabilized DE - ↑ Bioaccessibility and chemical stability of insulin and quercetin during digestion > water and W_1_/O - Lipid digestion: >80% for all samples, slower digestion for pectin-stabilized DE	[30]
- **Iron sulfate** (0.8%)	- Soybean oil - PGPR (5%)	- WPI (8–12%) - Tara gum (0–0.8) - Sucrose (0–2%)	20:80	20:80	- The smallest size (757.1 nm) at 12% WPI - Zeta potential: 44.53 mV to −49; 67 mV which to −42.73 mV to 41.3 mV after 7 days - The highest EE (96.95%) and storage stability (7 days) at 12% WPI, 0.8% tara gum, 2% sucrose - Digestion release: adults oral 8.36%, SGF 8.56%, and SIF 51.47%; infants SGF 27.22% and SIF 41.45% - Bioaccessibility: adults 49.54%; infants 39.71%	[15]
- High (1% and 1.5%)- and low-viscosity (1.5% and 2.4%) chitosan - Hydroxypropylmethylcellulose (HPMC; 1–1.8%) - **Vitamin C** (2%)	- MCT - PGPR (5%)	- SC (10%) - Sodium chloride (0.05–0.25%)	30:70	12.5:87.5	- Droplet sizes (6.0–6.4 μm) - Results of spray drying: Retention rate—98.0%- 101.5%; the highest EE (91.9%) at 2.4% low-viscosity chitosan. - Results of reconstituted emulsions: Retention rate of vitamin C—90% for all microcapsule; the highest EE (80.8%) at 2.4% low-viscosity chitosan; lower dissolution rate in SGF (35.9%) and SIF (44.5%) at 2.4% low-viscosity chitosan than control DE without hydrophilic emulsifiers during the first 10 min	[31]
- **Ferric sodium EDTA** (5%)	- Different oils including coconut oil and red palm oil - Different emulsifiers including Span 80 (20%), PGPR 90 (3–5%), ACETEM 90 (3–5%)	- SC (4%) - SA (2%)	40:60	21.7:78.3	- PGPR 90 presented the smallest internaland external droplet sizes - ↑ Particle size in red palm oil than coconut oil - Controlled delivery during digestion. - The highest EE (85.5%) for PGPR-stabilized DE - In vitro bioaccessibility: The highest iron release in SIF for DEs with coconut oil and red palm oil: ~72% and 75%, respectively - Higher iron release in SIF than in SGF	[32]
- Phosphoric acid buffer - **Vitamin C** (20 mg/mL)	- Soybean oil - PGPR (3%)	- Sipunculus nudus proteins (SPP; 0.5–2.5%) - **β-carotene** (1 mg/mL)	20:80	5:95 10:90 15:85 20:80	- ↓ Stability of emulgel at higher W_1_/O:W_2_ ratio > 10% - ↑ Gelling behavior and viscoelastic property with ↑ SPP - ↓ Particle size with ↑ SPP - EE at 1% and 2% SSP: 87.3% and 91.2% for vitamin C and 99.7% 99.8%, for β-carotene, respectively. - ↑ Antioxidant activity than the free bioactives	[14]
- **MgCl_2_** (0.125%)	- Sunflower oil - PGPR (2.5%) - Lecithin (2.5%)	- Lentil flour (LF, 15–30%)	40:60	40:60	- EE: inverse relationship with LF (at 15% and 30% LF, EE 97.54% and 92.42%); Higher EE by using a high-pressure homogenizer - Cake: lower hardness, more cooking loss, and equal volume of cake with DE compared to cake with free Mg - Protection: 79–83% of Mg preserved in DE after baking - Well-masked Mg flavor by DE	[16]
- **Vitamin B_12_** (0.2%) - NaCl (0.1 M)	- Olive oil - PGPR (8%)	- NaCl (0.1 M) - Different emulsifiers including Tween 20 (4%), soybean lipophilic protein (LP)-methyl cellulose (MC) (2–6%), LP (4%)	20:80	40:60	- Rheology: ↑ viscosity at 3:1 LP:MC ratio and 4:6 W_1_/O:W_2_ ratio - ↑ Emulsion activity and stability index at 3:1 LP:MC - EE: LP and LP-MC resulted in higher vitamin B_12_ EE than Tween 20; LP:MC ratio of 3:1 favored EE - Release: sustained GI release for W_1_/O to W_2_ ratio of 4:6 and LP:MC ratio of 3:1	[18]

### 2.2. Preservation Purposes

#### 2.2.1. Improved Antimicrobial Properties

Natural food compounds with antimicrobial characteristics hold significant importance for the future of the food industry and customer health. The encapsulation of these antimicrobial compounds, such as essential oils, natural extracts, and peptides, is considered an interesting approach to enhancing their physicochemical and microbiological stability as well as providing for their controlled release in the food matrix. In addition, the challenges related to the limitation of their consumption due to their possible interactions with other food ingredients, as well as their organoleptic taste, can be overcome by encapsulation in emulsion droplets [33,34]. Previous works have reported that the increased specific surface area of the emulsion-dispersed droplets can also improve the biological activities of essential oils and natural extracts; hence, lowering their required concentrations [35]. Therefore, several researchers have investigated the effect of DEs on the preservation effects of encapsulated antimicrobial agents, as evidenced in Table 2. Tessaro et al. [36] investigated the preparation of a “Pitanga” leaf hydroethanolic extract (PLHE)-loaded DE under the effects of tween 80 concentration (3–5%, and 8%) and the ratios of emulsion phases (30:70 and 40:60) [37,38]. According to antimicrobial activity measurements on PLHE, the PLHE diluted in the same concentration in the DE (PHLEd), and the most stable DE with a 40:60 W_1_/O/W_2_ ratio and 3% Tween 80, all samples showed an inhibition zone against *P. aeruginosa*, *Salmonella,* and *S. aureus* bacteria, but did not against *E. coli*. However, the antibacterial activity of PLHE (1.8–2.2 cm) was greater than that of the PLHE-loaded DE (0.6–0.7 cm). This fact could be related to the lower diffusion rate of encapsulated PLHE from W_1_ into the agar in disk diffusion tests as well as the possible reduction of extract during the emulsification process. In another study, the antibacterial properties of a honeybee pollen (HBP)-loaded DE against *Streptococcus pyogenes* were investigated using agar diffusion methods and serial dilutions [39]. The control HBP extract presented a lower inhibition zone (12 cm) compared to the encapsulated formulation (23 cm), suggesting its improved antibacterial activity. Interestingly, those DEs without HBP (only ethanol as the internal aqueous phase) also inhibited bacterial growth (22 cm), which might be due to the antimicrobial activity of chitosan hydrochloride in the formulation. The minimum inhibitory concentration MIC of the HBP-loaded DE was also significantly lower than its free form. In addition, the destruction of DEs using ethanol increased their antibacterial activity due to the higher release of HBP from internal layers of DE. Ji et al. [40] also developed DE-like intermediates through a dynamic encapsulation process by controlling the turbulent fluid flow at high temperatures to encapsulate hydrophilic nisin. The low inhibition-growth activity effect of nisin against *L. monocytogenes* under the effect of different shear stresses (12,500, 18,750 and 25,000 s^−1^) and temperatures (25–60 °C) confirmed the successful encapsulation approach to protect its antimicrobial activity as well as its sustained release in food systems. Moreover, microscopic images and surface observation recommended an even distribution of nisin within the polymer matrix and on the particle surface to enhance its activity. In addition, the microbiological aspect of gelled DEs incorporating perilla oil, a rich source of n-3 fatty acids, was studied in the presence or absence of hydroxytyrosol (Hyt) in W_1_ after 1 month of storage at 4 °C [41]. Microbial counts were relatively low at the initial time and the addition of Hyt resulted in the reduction in total viable counts by nearly 1 log. Bacterial growth increased significantly during storage in all formulations; however, it was delayed and lowered after the addition of Hyt during the first two weeks. The Hyt did not show positive inhibition activity on the growth of yeast and molds.

To conclude, encapsulation of antimicrobial agents in DE systems can enhance their microbial growth inhibition effect, increase the food’s shelf-life, and decrease microbial growth through the extended release of antimicrobial compounds without any interaction with the food matrix. Moreover, by changing the droplet size of the emulsion to smaller sizes, the larger ratio of surface area to mass can result in a unique interaction with microorganisms and host cells.

**Table 2 foods-13-00485-t002:** Overview of the past decade’s publications on enhanced antimicrobial properties of natural compounds encapsulated by W_1_/O/W_2_ double emulsions (DEs).

W_1_	O	W_2_	Phase Ratio		Remarkable Results	Reference
			W_1_:O	W_1_/O:W_2_		
- **Honeybee pollen (0.7–1.5 mL)**	- Lauroglycol 90^®^ (900–1200 mg), span 80 (360 mg), and Lipoid P75^®^ (60 mg)	- Chitosan (6–24 mg in 3.5 mL) - Pluronic F68 (230–700 mg in 3.5 mL)	-	-	- Droplet size: 90 nm - Zeta potential: +33 - EE ˃ 78% - ↑ Stability under SIF and storage - ↑ ORAC and antibacterial activity (inhibition zone) against *Streptococcus pyogenes* = 23 mm than free extract	[39]
- **Pitanga leaf hydroethanolic extract**	- Soybean oil - PGPR (3%)	- Tween 80 (3–8%) - SC (0.5%)	20:80	30:70 40:60	- Droplet size: 4.71–5.28 µm - Zeta: −30–−37.4 mV - Optimized formulation: 40/60 ratio and 3% Tween 80 - ABTS: 482 TE mg/g - FRAP: 2176 µmol TE/g - Inhibition zone against *E. coli*, *P. aeruginosa*, *Salmonella* ssp., *S. aureus*, were 0, 11, 8, and 13 mm, respectively	[36]
- Tween 80	- Corn oil - Span 20 - **Oregano essential oil** (170–680 ppm)	- Mixture of Tween 80 and Span 20 (13:87; 9.75%) - Inulin (3%)	40:60	20:80 30:70	- ↑ Antifungal activity against *A. niger* - Dose-dependent antifungal activity - The smallest droplet size during 20 days (2.68 and 3.05 μm) prepared at 20:80 ratio and 5800 rpm - The highest particle size and creaming at a 30:70 ratio at 2900	[42]
- NaCl (0.584%) - **Hydroxytyrosol** (Hyt; 0.125%)	- Perilla oil - PGPR (6%)	- SC (0.5) - NaCl (0.584%)	20:80	40:60	- Gelation with 4% bovine gelatine and 2% microbial transglutaminase—higher droplet size (3.72 μm) after Hyt addition than the control (2.55 μm) - Gel-like behavior, no frequency dependent properties, formation of weaker gels after Hyt addition - ↓ Hardness and chewiness and ↑ antioxidant activity after Hyt addition - ↓ Total viable count after 30 d at 2 °C	[41]

#### 2.2.2. Improved Antioxidant Activities

BHA, BHT, and TBHQ, as powerful synthetic antioxidants, are extensively utilized in fat-rich foods and emulsion-based formulations to inhibit or delay lipid oxidation. However, their safety is always a concern, resulting in a growing interest in natural antioxidant compounds to provide healthy diets [43]. Unfortunately, low water solubility, instability during processing and storage, and undesirable organoleptic characteristics, and poor availability, absorption, and permeability profiles make it challenging to formulate food products rich in these natural antioxidants [39]. As can be seen in Table 3, DEs offer the opportunity to deliver and protect various hydrophilic and hydrophobic antioxidants via different antioxidant mechanisms within different emulsion phases, potentially obtaining a synergic effect in food formulations. For instance, Silva, et al. [44] studied the lipid oxidation of DEs formulated with a blend of olive, linseed, and fish oils in the presence of gallic acid and quercetin in the internal and external aqueous phases under an accelerated condition at 60 °C for 1 month. The synergistic effect of both antioxidants resulted in lower hydroperoxide levels compared to control DEs, which was more noticeable after 12 days. Moreover, gallic acid and quercetin limited the possible formation of secondary oxidation products during mechanical stress and heating, as evidenced by a gradual increase up to nearly 21 days followed by a significant increase up to the end of the storage period. However, this process was much slower than the control sample. In another study, Ghiasi, Golmakani, Eskandari, and Hosseini [2] developed a structured PUFA-rich W_1_/O/W_2_ DE stabilized by gelation of the W_1_ and O phases and investigated its oxidation kinetics in the presence of gallic acid encapsulated in W_1_ and α-tocopherol in the O phase. They reported an extended induction period after the individual addition of both antioxidants (9–12 days), while the oxidation rate showed a significant reduction, confirming higher oxidative stability compared to the control DEs. However, α-Tocopherol offered a superior antioxidant effect, as evidenced by it producing higher values of antioxidant activity and stabilization parameters than gallic acid, due to its hydrophobic nature, resulting in its better location at the O/W_1_ interface as well as its strong placement at the gel state of the interface. Chaudhary, et al. [45] also investigated the chemical stability of different concentrations of a water-soluble extract of *Emblica officinalis* (EEO) encapsulated in the internal aqueous phase of a DE during storage at 4 °C for 63 days. The encapsulation efficiency exhibited a significant increase from 56.93% to 95–74% with increasing EEO concentrations from 15% to 50%; potentially due to the high binding capacity and hydrophilic character of the extract which resulted in a limited diffusion through the oil phase. According to the results of the DPPH, ABTS, and FRAP assays, the longer the storage time, the lower the antioxidant stability of DE. However, this reduction trend was significantly lower than the free EEO extract, suggesting a relatively strong protective effect of the DE structure. Eisinaitė, et al. [46] found the positive effect of black chokeberry pomace extract on the oxidative stability of the DE by increasing peroxide values and conjugating dienes less than the control DE without extract during convenient storage for 60 days. According to the results of DPPH^•^ scavenging, higher concentrations of BCPE exhibited powerful antioxidant activity after 21 days of storage due to the significant amounts of polyphenolics. Moreover, the control DE presented a higher rate of PV than pure oil due to the larger interfacial area of the emulsion droplets, resulting in more contact with the oxidants. However, the presence of encapsulated BCPE enhanced oxidation resistance, particularly at higher concentrations which could be related to the higher viscosity of W_2_ and hence decreases in the diffusion of prooxidants. Oxipres and Rancimat analysis in accelerated storage also confirmed that the BCPE in W_1_ can prolong the oxidation induction period remarkably and in a dose-dependent manner. In another study, Kumar and Kumar [47] reported lower TBARS values in reduced-fat meat batter formulated with a *Murraya koenigii* berry (MKB)-extract-loaded DE, as compared to DEs formulated with vegetable oil and animal fat. Interestingly, the MKB-loaded DE samples presented more oxidation stability in meat batter in comparison with a non-encapsulated extract, thus confirming the positive effect of DE structure on the prolonged antioxidative activity of bioactive compounds.

Therefore, the adverse effects of lipid oxidation in terms of developing off-flavors and loss of nutritional quality are always challenging, particularly in lipid-rich foods. Due to increasing customer demand for a healthy diet, the application of natural antioxidants is needed. In this regard, DEs can remove the limitations of using natural antioxidants, such as those with a high degradation rate during processing and storage, to provide higher oxidation stability. Moreover, DEs offer a chance to use the synergistic antioxidant effect of both polar and non-polar natural antioxidants by co-encapsulation. Additionally, by modulating the droplet size (interface area) and location of the added antioxidant agent in DE systems, the oxidation rate of foods can be decreased considerably.

**Table 3 foods-13-00485-t003:** Overview of the past decade’s publications on enhanced antioxidant properties of natural compounds encapsulated by W_1_/O/W_2_ double emulsions (DEs).

W_1_	O	W_2_	Phase Ratio		Remarkable Results	Reference
			W_1_:O	W_1_/O:W_2_		
- **Hyssop extract**	- Soybean oil - Span 80 (25%)	- Different emulsifiers including soy protein isolate (SPI; 3%)/chia seed gum (0.1%) - SPI (6%)	7:93	30:70	- ↑ EE with SPI/chia seed gum (87.69%) than SPI (80.71%) - Shear-thinning behavior - Oxidative stability: ↑ PV and p-Anisidine time-dependently; lower PV and p-Anisidine with SPI/chia seed gum than SPI and free extract - Higher zeta potential SPI/chia seed gum (31.533 mV) and smaller droplets (190.833 nm) than SPI	[48]
- **Gallic acid** (200 ppm) -NaCl (100 mM) - Fe_2_(SO_4_)_3_ (5mM) - In gelled emulsion:κ-carrageenan (1%), KCl (100 mM)	- Linseed oil - PGPR (7.5%) - Monoglyceride (7.5%) - **α-tocopherol** (1045 ppm)	- Tween 80 (4%)	20:80	40:60	- ↓ D4,3 and ↑ stability after O phase gelation - ↑ Negative values of zeta potential for all formulations - Rheology: weak frequency-dependent and higher viscoelastic behaviors after O phase gelation - Higher values of PV, TBAR, p-Anisidine, and conjugated diene during storage, especially after O gelation - ↑ Induction period after antioxidant addition, higher effectiveness of α-tocopherol than gallic acid based on polar paradox	[2]
- **Black chokeberry pomace extract (BCEP)** (15–35%) - NaCl (0.5%)	- Rapeseed oil - PGPR (3%)	- Milk protein solution (14%)	20:80	30:70	- High thermal stability (100–91.2%) at 4 °C for 60 days - D_4,3_: ranged from 61.66 μm to 37.65 μm, ↓ D_4,3_ at higher concentration of BCEP and during storage - ↑ Viscosity at higher concentrations of BCEP - High EE (>95%) during storage - ↓ DPPH^•^ scavenging after 21 days of storage at 4 °C from 36.32–44.50% to 12.12–15.40% - ↑ DPPH^•^ scavenging at higher concentrations of BCEP - ↑ PV six to ten times lower (4.86–7.47) than control DE (47 meq O_2_ kg^−1^) during storage at 37 °C for 60 days - ↑ Conjugated dienes and trienes during storage at 37 °C for 60 days less than control DE	[46]
- **Brassinolide (0.008%)** - Gelatin (1%) - NaCl (0.1%)	- Olive oil - **Cinnamon essential oil (2.66%)** - PGPR (1.66%)	- Different emulsifiers including whey protein concentrate (WPC)-gum arabic and WPC - high methoxyl pectin (HMP) at 1:3; 1:1; 3:1 ratio	10:30	10:30	- Optimized formulation: DE stabilized by WPC-HMP (1:3) with largest particle size (581.30 nm), lowest PDI (0.23) and zeta potential (−40.31 mV), and highest EE of brassinolide (92%) and cinnamon essential oil (88%). - Results of broccoli coating: ↑ Chlorophyll content and ↓ activities of chlorophyllase (9%) and magnesium-dechelatase (24%), and a lower rate of respiration after storage than control broccoli - Activated energy metabolic enzymes (SDH, CCO, H+-ATPase, Ca2+-ATPase), ↑ ATP, and energy charge.	[49]
- **Murraya koenigii berries extract (MKB)** - NaCl (0.6%)	- Soybean oil - PRPG (6%)	- WPC (6%) - NaCl (0.6%)	50:50	70:30	- ↑ Emulsion stability, cooking yield, hardness and lightness of meat batter - ↓ Shrinkage and redness values of meat batter - ↑ *G′*, *G″*, and η* of meat batter - The order of TBARS during storage at 4 °C for 9 days: meat batter with vegetable oil > animal fat > control DE > DE and free MKB > MKB-loaded DE - ↑ Oxidation stability of lipid phase and meat batter	[47]
- ***Emblica officinalis* (EEO) extract** (15–50%) - NaCl (1–2%)	- Rice bran oil - PGPR (2–4%).	- Different emulsifiers (0–4%) including low methoxy pectin (LMP), gum Arabic, WPC, SC	30:70	30:70	- Optimized DE: 2% NaCl, 50% EEO, 4% PGPR, 2% LMP - D_4,3_: 72.95 µm, high EE (>90%), ↓ during storage - Viscosity: 0.715 Pa.s, ↓ during storage - Zeta potential: −32.17 mV, ↓ during storage - ↑ Encapsulation efficiency at higher EEO concentration - ↑ Antioxidant stability than free extract - ↓ Antioxidant stability after 3 months at cold storage less than free extract (↓ ABTS of control and DE from 7872 and 7473 to 753 and 2969 mM TE g^−1^, respectively)	[45]
- Catechin (750 µg/mL) - Gelatin (3%)-NaCl (2%) - **Ascorbic acid** (0.2%)	- Olive oil - PGPR (6%) - **Curcumin** (0.1%)	- Tween 80 (1%) - Ascorbic acid (0.2%) - NaCl (2%)	25:75	25:75	- D_4,3_: ↓ from ≈3.88 for the blank to ≈2.8–3.0 µm for curcumin and/or catechin-loaded as well as co-delivery DE - Zeta potential ≈ −20 mV - EE: 88% for curcumin, 97% for catechin, >80% for co-delivery DE. - Loading efficiency: 0.1% for curcumin, 0.075% for catechin, 0.175% for co-delivery DE - In vitro release: controlled release of curcumin from curcumin- and co-loaded DE, burst release of catechin from catechin- and co-loaded DE (≈30% within 30 min and >45% within 1 h) - In vitro bioaccessibility: ≈72% for curcumin-loaded, 68% for curcumin in co-delivery DE, ≈16% for free curcumin, ≈60% for catechin-loaded, ≈54% co-delivery DE, and ≈10% for free catechin	[50]

The * in η* is a symbol of complex viscosity.

### 2.3. Protection of Enzyme Activity

Enzymes are multipurpose biological molecules with extensive applications in food, pharmaceutics, and medicine. To date, different approaches have been investigated to enhance their environmental sensitivity and low bioavailability during processing, storage, and oral administration including genetic engineering approaches, immobilization, and encapsulation [51,52]. In this regard, DEs can be considered an efficient tool for the protection of enzymes and hence improve their stability and activity (Table 4). For instance, Li, et al. [51] designed a DE delivery system stabilized by complexes of soybean protein isolate (SPI) and polyglutamic acid (PGA) at different volume ratios of 5:1, 3:1, 1:1, 1:3, and 1:5 for encapsulation of nattokinase (NK). The highest value of EE (97.19%) for hydrophilic NK was found for DE coated with a 1:3 ratio complex compared to individual SPI and PGA coatings. The fact was related to the increased exposure of lipophilic amino acids and hydrophobic interactions, resulting in the superior stability of the colloidal network in DE. Similarly, the highest NK bioavailability (80.69%) was observed for DE coated with a 1:3 ratio complex, while PGA-stabilized DEs, with their loose emulsified structures, had the lowest bioavailability. Indeed, SPI-PGA complexes presented a controlled release profile due to the PGA gelation, obtaining a delayed oil phase release. However, a higher PGA ratio affected the hydrolysis of the interfacial layer and reduced the release of NK from the W_1_ phase. Similarly, Wang, et al. [53] developed nattokinase-loaded DE systems under the effects of different concentrations and compositions of W_1_: a lipid phase type and emulsifier type. Sustained release behaviors of the nattokinase-loaded DE at optimized emulsification conditions in four media, including water, phosphate buffer solution, HCl solution, and acetic acid-sodium acetate buffer, were confirmed, with the highest and lowest values obtained at pH 1.21 and 6.8, respectively. The high encapsulation efficiency (86.8%) showed a good protection effect of optimized DE formulation. In addition, pharmacodynamics evaluation revealed the effective prolog effect of the encapsulation of nattokinase in the DE on the whole blood clotting time in mice as well as enhancing mouse tail thrombosis in comparison with normal saline and nattokinase solution, suggesting the potential application of DEs for nattokinase protection against a gastric acid environment and hence provide an enhanced efficiency during oral administration. In another study, Souza, et al. [54] microencapsulated microbial lactase (originated from *Aspergillus oryzae* and *Kluyveromyces lactis*) in DEs followed by complex coacervation. The best microcapsules at the optimum concentration ratio of core solution-to-total polymer (1%) presented low a_w_ (≤0.4) and particle size (≤93.52 μm), and high EE (≥98.67%). Significant enhancements of pH stability, storage stability, and temperature stability for the encapsulated lactase were reported as compared to the free form. Moreover, the encapsulation of lactase in a DE presented a low release rate (10–20%) in simulated gastric fluid (SGF) and a high release rate (>80–95%) in SIF, mainly due to the effects of pH, pancreatic activity, and bile salt, to improve the demulsification of the microcapsules.

In summary, DEs can be introduced as successful carriers to increase enzyme stability against different environmental conditions, resulting in higher bioavailability, prolonged activity, and controlled target release as well as providing their recycling capacity.

### 2.4. Improved Viability of Probiotics

Probiotics are a group of live bacteria that confer health beneficial effects on their host. To exert their effects on the human body, certain concentrations (more than 10^6^–10^7^ CFU/g or CFU/mL) of the probiotics should reach the colon; however, the susceptibility of the microorganisms during processing and storage, as well to the acidic nature of some foods and gastrointestinal (GI) conditions, makes it challenging to prepare probiotic-loaded food products. To sustain the viability of probiotics, encapsulation in W_1_/O/W_2_ DEs has been suggested to protect the cells against environmental conditions [55,56]. Table 5 presents recent successful applications of DEs for probiotic encapsulation. In this regard, Frakolaki, et al. [56] successfully encapsulated *Bifidobacterium animalis* subsp. Lactis BB-12 within a W_1_/O/W_2_ DE using the extrusion technique. The probiotics were added to the W_1_ and then extruded using encapsulating materials. The cell viability was examined during storage and through passing SGF. They reported the highest viability of BB-12 cells encapsulated within DE at 4 °C after 1 month of storage (>10^6^ CFU/g) compared to those encapsulated through conventional extrusion and those within the W_1_ of DE before extrusion. The survival rate of the cells within the DE was also higher (68.6–86.1%) than the cells encapsulated through a conventional extrusion (48%) during passing SGI. Exposure to different pH values in the range of 4–8 showed higher viability at pH 7–8. However, BB-12 cells encapsulated in the DE had more than 80% viability even at low pH values (compared to the 67.32% cell viability in the conventional extrusion), indicating its high efficiency in protecting the probiotic cells. Marefati, et al. [55] used a DE to encapsulate *Lactobacillus reuteri* within the internal aqueous phase. They reported that storage of the DE samples at 6 °C did not affect the cell viability after 3 days, and then the cell count decreased by 5.2 and 2.82 log CFU/mL on the 15th and 30th days of storage, respectively. However, the control sample (the probiotics encapsulated within the outer phase (W_2_) of the DE), showed a decreasing trend in the cell count from the 1st day and reached 0 after 30 days. Investigation of the cell viability in simulated GI condition showed a survival rate of 70% for encapsulated *L. reuteri* and 2.8% for the free cells (control sample) under gastric conditions. Under intestinal conditions, the encapsulated cells showed a gradual reduction and reached 4.46 log CFU/mL from 6.53 CFU/mL after 180 min; however, the free cells count had a sharp reduction under intestinal conditions and decreased from 4.7 to 2.69 log CFU/mL in 180 min. This study indicated the role of W_1_/O/W_2_ DE in protecting probiotics in GI until reaching the colon. In another study, by Qin, et al. [57], a pH-sensitive DE was designed to encapsulate and colon release *Lactobacillus plantarum*. In this study, the DE was stabilized by WPI-epigallocatechin gallate (EGCG) conjugate particles and pH-sensitive alginate-Ca-EDTA was added to the W_1_. It was observed that the free probiotics count experienced a steep decrease under the simulated GI conditions, especially under the gastric conditions (from 7.81 × 10^7^ to 0.14 × 10^7^ CFU/mL); however, the encapsulated probiotics (in W_1_) experienced a small count reduction under the gastric conditions due to the resistance of the Ca-alginate hydrogel to the acidic conditions and pepsin. In the SIF, the middle oil phase (medium-chain triglycerides (MCTs)) protected the cells against bile salts and digestive enzymes and, hence, a small loss occurred (from 7.79 × 10^7^ to 7.39 × 10^7^ CFU/mL). It was shown that the DE at a 3% WPI-EGCG particle concentration, 0.6% MCT oil concentration, and 3% PGPR resulted in the highest protective effect against GI conditions and can be used as a colon-targeted release vehicle for probiotics. In a study by Jiang, et al. [58], *Lactobacillus acidophilus* was encapsulated in a W_1_/O/W_2_ DE. It was observed that the cell viability of the encapsulated and free probiotics in SGF was 85.1% and 9.8%, respectively, and, in SIF, the encapsulated cells had a 70.8% viability (4.95 × 10^6^ CFU/mL), while the free ones showed 7.8% (3.86 × 10^5^ CFU/mL). The effect, on the viability of the probiotic cells, of adding fish oil and SA to the oil phase and the outer aqueous phase, respectively, was assessed. It was concluded that fish oil can promote cell growth and enhance the DE’s protective effect against GI, and SA could reduce the cell release in simulated GI conditions by creating a layer around W_1_/O. In another study, the viability of *Lactobacillus plantarum* in W_1_/O/W_2_ DE-loaded alginate capsules under simulated GI conditions was examined and compared with the viability of free cells and cells loaded in the DE without alginate capsules. The free cells underwent a 3.9 log cycle reduction under the gastric conditions (120 min) and then further reduction occurred after incubation under intestinal conditions (a 5.79 log cycle reduction). Remaining under the intestinal conditions for 2 more hours did not change the cell count. The reduction of encapsulated *L. plantarum* in the DE under gastric conditions was only 1.02 log cycle and the DE structure remained intact; however, it was destroyed under the in vitro intestinal conditions and caused a considerable decrease in the cell count. The DE-loaded alginate capsule provided the highest protection against GI conditions (a 0.81–1.19 log cycle reduction in the gastric and a 3.04–3.22 log cycle reduction in the intestinal conditions, depending on the time needed for Ca gelation). The best result for this sample appeared after 20 min of gelation with Ca. The storage stability of the encapsulated cells in DE after 42 days of storage at 37 °C indicated a gradual decrease in the cells until the 21st day and then a sharp decrease until the end of storage (survival rates of about 95% and 62% in the 7th and 42nd days of storage were reported, respectively) [59]. Abbasi, et al. [60] investigated the viability of *L. plantarum* encapsulated in the inner phase of a W_1_/O/W_2_ DE under the effects of heating and pH in the presence of various gelling agents (gelatin, alginate, tragacanth gum, and carrageenan) in W_2_. They reported that the initial count of free cells (9.95 log CFU/mL) reached 9.97, 8.10, 6.37, and 0 log CFU/mL after 2 min of heating at 30, 50, 63, and 72 °C, respectively. The control sample (encapsulated cells in a DE without any gelling agent) also showed a significant decrease from 9.95 log CFU/mL to 5.47 log CFU/mL after heating at 72 °C. However, in the presence of the gelling agents, the probiotics were better protected against heat and their count remained constant (in the range of 6.69–7.33 log CFU/mL after heating at 72 °C). The cell viability at different pH values (2, 3, 6.5, and 7) showed the highest cell reduction at a pH of 2 for all samples with the lowest viability seen in the free cells and the highest in those encapsulated in DE in the presence of carrageenan. It was concluded that gelation of the outer phase could increase the viability of the cells and carrageenan was introduced as the most efficient gelling agent to enhance viability. The authors declared that, in addition to creating a physical barrier, the positive role of carrageenan was related to its prebiotic activity. Silva, et al. [61] prepared a sweet mango dessert containing *L. plantarum* encapsulated in the W_1_ of a DE and then studied the cell viability of the probiotics under in vitro digestion. The measured numbers of probiotics encapsulated in the DE and those dispersed directly in the formulation of mango candy were 7.92 and 7.61 log CFU/mL at time 0, respectively, and did not decrease during the emulsification process. The presence of glucose and lactose in the formulation promoted the growth and viability of the cells. Exposing to GI condition showed that DE could provide good protection to the cells against SGF; however, the protection was not enough against the intestinal conditions and the cell count decreased. The free cells survived in the gastric condition and their count was close to that of the encapsulated ones in the SIF.

It is concluded that the encapsulation of probiotic bacteria in a W_1_/O/W_2_ emulsion can be considered a novel encapsulation approach that can enhance their viability during storage and retard their release under GI conditions without requiring sophisticated equipment. However, to achieve higher cell availability and provide higher levels of protection, additional treatments such as gelation of the aqueous phase/phases and adding prebiotics to W_1,_ are recommended.

### 2.5. Improved Sensory and Color Attributes

The sensory properties (e.g., texture, appearance, and flavor) of a food emulsion are influenced by the initial emulsion characteristics (i.e., the properties of the continuous phase, dispersed phase, and interfacial region). The behavior of the food emulsion during mastication also plays an important role in its sensory properties as a result of the changes in its structure and composition brought about by saliva dilution, chewing, and surface coating. The term ‘flavor’ is an integrated response of taste, aroma, texture, and mouthfeel. Therefore, in addition to the aroma and taste, mouthfeel and texture play key roles in the perceived flavor. The optical properties of (multiple) emulsions, including color and opacity, are affected by the microstructure and composition (i.e., particle size, particle concentration, and refractive index contrast) [63]. The overall flavor properties of emulsion systems are influenced by the distribution of the flavor molecules within different phases (e.g., water, oil, interface, headspace), and their release behavior during consumption. The release of the flavor molecules from food emulsions is evaluated by their mass transport kinetics and their equilibrium partition coefficients. In W_1_/O/W_2_ DE systems, the middle oil phase is a filler that does not contribute to taste but affects it in two different ways: (1) Oil droplets increase the concentration of taste components in W_2_. At a constant concentration of a taste compound, increasing the oil fraction in the DE leads to increasing the taste compound concentration and consequently more intense precipitation. And (2) the oil, as a barrier in the middle phase between the two aqueous phases, can prevent the migration of a taste compound in W_1_ from reaching W_2_. As in intact DE systems, only W_2_ touches the taste buds on the tongue surface, the W_1_ in the DE can be used to encapsulate unpleasant taste compounds to mask them during oral consumption [64]. Table 6 shows the potential applications of DEs in improving sensory characteristics. Polypeptides derived from the enzymatic hydrolysis of proteins have positive physiological and health beneficial effects. However, their bitter taste is the main limiting parameter for their consumption. Among different encapsulation methods suggested for bitter polypeptides (PBs), using a W_1_/O/W_2_ DE seems to be one of the most appropriate ones. Gao, et al. [65] encapsulated PBs in the W_1_ of DE and then examined the effect of adding gelatin (as a gelling agent) at 1% and 2% concentrations to W_1_, W_2,_ or both aqueous phases of the DE, on the DE’s properties and BP release (DE-1: DE without gelatin, DE-2: 2% gelatin in W_1_, DE-3: 2% gelatin in W_2_, DE-4: 2% gelatin in both W_1_ and W_2_, and DE-5: 1% gelatin in both W_1_ and W_2_). It was observed that the presence of gelatin could decrease the size of the DE droplets, especially when it was added to both W_1_ and W_2_ (D_3,4_: DE-1 178.3 µm, DE-4 9.38 µm). In addition, 2% gelatin was more efficient in terms of size reduction than 1%. Gelatin addition also increased the viscosity and physical stability of the DE samples (DE-1 < DE-2 < DE-3 < DE-5 ≤ DE-4). The encapsulation efficiency (EE) of BPs was significantly increased after gelatin addition. Gelation in W_1_ had a more predominant effect in this regard and, hence, the EE of DE-3 was lower than the other gelled samples. Assessing masking the bitter taste by DE showed the ability of DE to mask bitterness in the presence of gelatin and the higher importance of the gelation of W_2_ than W_1,_ with no significant differences between the 1% and 2% gelatin concentrations. In vitro release of BPs in a dialysis bag proved that the gelation of W_1_ or W_2_ could retard BP release (DE-1 < DE-3 < DE-2 < DE-5 < DE-4). In vitro digestion of DE-4, as the best sample, was performed under simulated GI conditions and indicated that the DE sample remained intact in the oral stage and resulted in a very low bitterness in terms of sensory evaluation. Under the gastric conditions, the DE protected the BPs from the acidic conditions and enzymes and, finally, released them under the small-intestinal condition (as the desired site of release). The simultaneous inoculation of *Zygosaccharomyces rouxii* and *Tetragenococcus halophilus* in the internal W_1_ and external W_2_ phases of W_1_/O/W_2_ DEs with reduced NaCl and/or substitution with KCl in the moromi stage of soy sauce fermentation was carried out by Devanthi, et al. [66]. In the presence of the 18% NaCl, the growth of *T. halophilus* was stopped during the first two weeks, while partial substitution of NaCl (6% NaCl and 12% KCl) promoted its growth and enhanced lactic acid production. However, the final aroma was different from that obtained in the presence of 15% NaCl (original method), and lower amounts of alcohols, acids, esters, furan, and phenol were detected. At a reduced salt concentration, *T. halophilus* grew faster when it was simultaneously incubated with *Z. rouxii* compared to sequential incubation and, by producing volatile components such as some alcohol and ester types under simultaneous incubation, could compensate for the low-salt conditions, making it possible to produce soy sauce at low-salt concentrations with an aroma profile close to that achieved using the original method. This study demonstrated the effective application of DEs for delivering the mixed cultures in low-salt soy sauce without compromising its quality. Jamshidi, et al. [67] used a DE to encapsulate fish protein hydrolysate and fish oil within the W_1_ and the middle oil phase, respectively, to mask the unpleasant flavor and taste. After optimizing the DE formulation, it was freeze dried and added to yogurt for fortification. Sensory analysis of the fortified and control yogurts showed no differences between the homogeneity before consumption (appearance) and in the month, and samples did not have any rancid or bitter taste after fortification. However, the high concentration of fish oil caused a fishy taste that can be masked more efficiently by using flavoring agents. Buyukkestelli and El [64] used a DE to enhance the sweetness of saccharose and formulate reduced-sugar foods. They designed a DE at a 16:24:60 (W_1_:O:W_2_) ratio and a single emulsion (control) at a 40:60 (O:W) ratio. Saccharose at a 15 g/100 g concentration was added to the outer phase of each emulsion. The results of the sensory evaluation to assess and compare the sweetness of the two samples revealed more intense sweetness for the DE despite the same saccharose concentration being present in both samples. In fact, the saccharose concentration that contributed to perceptions of the flavor and interacted with the taste buds was 25 g/100 g for the DE and 19.74g/ 100 g for the control. In a study by Chen, et al. [68], the gelling of the middle oil phase of a Pickering W_1_/O/W_2_ DE by beeswax (BW) was used to control the release of aroma (2,3-diacetyl) from W_1_. In this study, the effects of different BW concentrations (0–8%), temperature (25 and 37 °C), and time (7 days) on aroma release were examined. It was shown that by increasing the BW content the flavor release considerably reduced. At a BW concentration of 0%, there was no difference between flavor release at 25 °C and 37 °C. At 2% and 4% BW, a greater extent of flavor release occurred at 37 °C due to the partial melting of fat crystals that facilitate the aroma release. At 6% and 8%, aroma release was not significantly different at the two temperatures due to the high compactness of the fat crystals. Investigation of aroma release during the initial storage period showed a burst release in the absence of BW (51.04% aroma decrease). Gelation of the oil phase caused an extended aroma release and the smallest decreasing amplitude (28.21%) was recorded for 8% BW, showing the slowest release due to the steric barrier of the fat crystals and the higher strength of the gelled oil phase at this concentration.

On the other hand, the color of foods is also considered an important parameter in achieving visual attraction to improve consumer appetite and overall acceptance. Encapsulation by DEs is an interesting method to improve the physicochemical stability, bioaccessibility, and controlled release of food colorings. Table 6 lists potential applications of W_1_/O/W_2_ emulsions in improving color stability and hence the biological properties of natural pigments. To this end, Nunes, et al. [69] compared the potential of applications of DEs to enhance the stability and bioaccessibility of lutein with a single O/W emulsion. They designed three different DEs, W_1_/O/W_2_ without lutein, W_1_/O-L/W_2_ containing lutein in the oil phase, and W_1_-L/O/W_2_ containing lutein in the WPI nanoparticles in W_1_ during the desolvation process. Lutein content in all fresh samples was in the range of 18.3–19.9 µg.g^−1^ which decreased to 11.3–12.4% after 14 days of storage. The DEs showed higher EE (>99%) than the O/W emulsion (94%). The highest (43.1%) and the lowest (31.8%–34.3%) lutein losses under LED sunlight lamps were reported for W_1_-L/O/W_2_ and both W_1_/O_L_/W_2_ and O/W, respectively. This was related to the protective effect of oil and/or PGPR in terms of the chemical stability of lutein. Color stability during storage confirmed the better potential of DEs than SEs, as evidenced by more yellow color. According to an in vitro digestion study, lutein recovery (98.7–99.9%) and bioaccessibility (20.8–28.2%) for DEs were significantly higher than the W/O emulsion (90.3% and <10%), suggesting that it is less prone to degradation during digestion after being incorporated into DE structures. In another study, Li, et al. [70] investigated the color changes of anthocyanin-loaded DEs as affected by temperature storage. Their results represented increases in lightness, yellowness, and ΔE values, and a decrease in redness in all anthocyanin after storage at 4 °C and 25°C for 28 days in a dark place. However, changes in the color parameters at 25 °C were significantly higher than 4 °C due to the higher sensitivity of both emulsions and anthocyanin under higher temperatures. The greater yellowness and less reddishness were attributed to the reduction in flavylium cation and the hydrolysis of a double bond in the anthocyanin molecule. Moreover, the EE of anthocyanin decreased from 95.3% in the fresh samples to 93.2% and 88.9% at the end of storage at 4 °C and 25 °C, respectively. This was mainly due to the anthocyanin leakage from W_1_ to W_2_ by osmotic pressure, and the higher temperature could accelerate this driving force. Tang, et al. [71] also introduced Pickering DEs stabilized by sugar beet and pectin-bovine serum albumin nanoparticles as a reliable delivery system for improving the stability and bioaccessibility of betanin and curcumin. The blue appearance of control DE changed to purplish-blue in betanin-loaded DE and green in curcumin-loaded DE. No color change was observed for all DE samples during storage at 25 °C for 3 and 7 days. In contrast, the color of free betanin progressively faded and degraded to brownish as a result of light excitation and nucleophilic attack. Similarly, free curcumin changed gradually from brilliant yellow to greenis-yellow. After 7 days of storage, the content losses of free betanin and curcumin were 56.5% and 42.1%, respectively; while incorporating DEs increased the retention of betanin and curcumin to 76.6% and 81.5%, respectively. Moreover, the EE values of curcumin and betanin were 84.1% and 65.3%, respectively. Meanwhile, the loading efficiency (LE) of betanin was more than that of curcumin. The DEs strategy also remarkably increased the bioaccessibility of free betanin and curcumin after digestion from 15.6% and 23.1% to 42.7% and 53.5%, respectively. Therefore, the susceptibility of the encapsulated colorants to acidic pH was decreased by incorporation into DEs. Co-delivery of betanin and curcumin by DEs also synergistically limited the growth of cytotoxic A549 cells. In addition, Yu, et al. [13] successfully developed a novel design of pH-sensitive DE-filled hydrogel containing hydrophobic astaxanthin and hydrophilic phycocyanin by adding gellan gum. The visual appearance of the DEs was yellow due to the brown color of the astaxanthin. Indeed, the blue color of phycocyanin was shielded due to its encapsulation in W_1_. The EE was increased by increasing gellan gum concentrations and the maximum values for astaxanthin (90.82%) and phycocyanin (94.1%) were achieved at gellan gum concentrations of 0.5% and 0.7%, respectively. The DE network also enhanced the storage stability of astaxanthin and phycocyanin as evidenced by there being no color changes after 10 days of storage, while the apparent yellow color of the astaxanthin solution and blue color of the phycocyanin solution faded, indicating their rapid breakdown. The degradation rates of the free form of astaxanthin and phycocyanin were 33.2% and 23.4% after 3 days, which increased to 79.9% and 48.1% after 10 days. After encapsulation in DEs, the degradation rates of astaxanthin and phycocyanin decreased to 25.6% and 16.8% after 10 days, respectively. According to in vitro study, the DEs efficiently protected the phycocyanin and astaxanthin during gastric digestion by decreasing their release rates (<25%), which reached more than 60% after digestion in the small intestine. Moreover, the low values of bioaccessibility for the free phycocyanin (12.54%) or astaxanthin (14.27%) in water increased significantly after encapsulation in DE and DE emulsion gels due to the formation of mixed micelles.

Consequently, DE offers a simple and easy-to-handle approach to both masking and enhancing flavors. The encapsulation of a compound in W_1_ can mask its aroma. Moreover, the gelation of each of the three phases of a DE can retard releasing the compound. Generally, the addition of a compound to W_2_ of a DE can enhance its flavor compared to the same concentration of it in the aqueous system. In addition, a W_1_/O/W_2_ DE offers an efficient carrier system to encapsulate and protect natural colorant agents in food systems with increased stability against environmental conditions and harsh processing and to improve their bioaccessibility, as well as offering the possibility of controlled release.

**Table 6 foods-13-00485-t006:** Overview of the last decade’s publications on enhanced antimicrobial properties of natural compounds by W_1_/O/W_2_ double emulsions (DEs).

W_1_	O	W_2_	Phase Ratio		Remarkable Results	Reference
			W_1_:O	W_1_/O:W_2_		
- **NaCl** (0.25–1%)	- Sunflower oil - PGPR (6%) - Monoglyceride (8%)	- Modified starch (4%)	20:80	40:60	- ↑ EE by oil gelation - The results of preparation of a low-salt burger: - ↓ 25% salt by replacing DE in the burger with desirable saltiness ↑ Antioxidant stability and ↓ color changes during storage in the presence of cinnamaldehyde in the oil phase - ↓ Cooking loss by adding DE - Undesirable changes in textural properties by adding DE	[72]
- **Bitter peptide** (50%) - Tartrazine (2%) - Gelatin (0–2%)	- Palm oil - Camellia oil - PGPR (4%)	- Gelatin (0–2%) - SC (2.5%)	40:60	40:60	- Size range: 9.38–52.3 µm - ↑ Gelatin led to ↓ size - ↑ Viscosity and ↑ physical stability by gelatin - EE > 80% by gelation - ↑ EE and ↓ release by W_1_ gelation - ↑ Bitter taste masking by W_2_ gelation - ↓ Peptide release ↑ gelatin	[65]
- **2,3-diacetyl** - Citric acid - Sodium sulphate buffer - SA(0.5%)	- Soybean oil - Beeswax (BW, 0–8%) - PGPR (2%)	- Bacterial cellulose (1%)	30:70	50:50	- ↑ Size by ↑ BW up to 4% - *G′* < *G″* at BW 2–4% - ↑ BW led to ↑ stress at the crossover point and shear thinning behavior - ↑ BW led to ↑ viscosity and ↑ friction coefficient - ↑ BW led to ↓ ∆BS and ↓ thickness, ↓ osmotic pressure and ↑ stability at 6–8% - BW 6–8% led to ↓ aroma release with no difference at 25 and 37 °C, extended aroma release by BW (smallest decreasing amplitude for 8%)	[68]
- MgCl_2_ (5%) - Na-caseinate (0.5%)	- Olive oil - PGPR (1–6%)	- **Saccharose** (≤15% of total DE) - Na-caseinate (12.5%)	40:60	40:60 35:65 30:70	- Smaller droplets at 40:60 (8.58 µm) - Shear thinning and pseudoplastic behavior (n < 1) - The lowest viscosity at 40:60 - The lowest hydrolysis degree (of oil) in GI at 35:65 - ↑ Sweetness of DE (40:60) compared to O/W by closely 75%	[64]
- **Mg_3_(C_6_H_5_O_7_)_2_** (0.025 M) - **MgSO_4_** (0.075 M) - **MgCl_2_** (0.075 M) - **Mg(C_3_H_5_O_3_)_2_** (0.075 M) - **Mg(C_3_H_5_O_3_)_2_** (0.075 M) + **Lactose** (0.225 M) - **CaCl_2_** (0.075 M) - **NaCl** (0.075 M) - **CsCl** (0.075 M) - **CsCl** (0.15 M)	- Miglyol oil - PGPR (5%)	SC (3.1%) Lactose (0.125 to 3 M)	40:60	10:90	- Anion and cation affect the release rate of salts - ↑ Release of each encapsulated salt from W_1_ to W_2_ during storage - ↑ Complexation constant of Mg^2+^ with its counterion led to ↓ the release - ↓ Hydration enthalpy of Mg^2+^ counter ion led to ↓ the release - ↑ Release rate for chloride ions in case of monovalent ions (Cs^+^ and Na^+^) than divalent ions	[73]
- **Fish protein hydrolysate** (12%) - NaCl (4%) - Vitamin B12 (4%)	- **Fish oil** - PGPR (6–10%)	- WPC - Inulin - WP/inulin ratios: 1/1, 1.608/1, 2.5/1, 3.39/1, 4/1	30:70	50:50 61.65:38.34 71.47:28.57 77.22:22.78 80:20	- The optimum parameters: 2:1 ratio of wall/core, 2.12:1 ratio of WPC/Inulin, and 6.28% PGPR ↓ WPC/inulin and ↑ PGPR led to ↓ release of the vitamin - ↓ WPC/inulin, ↓ W1O/W2, and ↑ PGPR led to ↓ creaming - ↓ WPC/inulin and ↑ PGPR led to ↑ EE - Interaction between W1/O/W2 and WPC/inulin led to ↓ encapsulation stability of the vitamin - WPC/inulin and W1O/W2 led to positive and negative effects on the encapsulation stability of the vitamin - Results of fortified yogurt with DE - Optimized DE condition for sensory analysis led 2:1 mass ratio of W1/O to W2, 2.12:1 ratio of WPC to inulin, and 6.28% PGPR The addition of a flavoring agent was recommended	[67]
- **NaCl** (6%) - *Z. rouxii* (10^5^ CFU/mL)	- Soybean oil - PGPR (2%)	- NaCl (6%) Tween 80 (1%) - *T. halophilus* (10^6^ CFU/mL)	20:80	20:80	- The results of reduced NaCl and/or substitution with KCl in soy sauce fermentation: - Non-newtonian behavior - ↑ Brine to koji led to ↓ viscosity - ↓ Size during fermentation (27.88 to 11.40) - ↓ Size by adding DE to moromi - Partial NaCl substitution led to ↑ *T. halophilus* growth and ↑ lactic acid - ↓ NaCl led to ↑ *T. halophilus* growth to 8.88 log CFU/mL, faster sugar depletion, and ↑ lactic acid production. - Simultaneous incubation led to ↑ *T.halophilus* growth and production of volatile compounds	[66]
- Gelatin (10%) - NaCl (0.4%)	- Sunflower Oil - PGPR (4–9%)	- WPI (1%) - NaCl (0.2%)	30:70 40:60 50:50	30:70 50:50	- 70% W_2_ and ↓ oil content (↑ % fat replacing) led to ↓ droplet size, ↓ viscosity, and ↑ expelled gelatin in W_2_ - 50% W_2_ and ↓ oil led to ↑ expelled gelatin in W_2_, ↑ viscosity and ↑ yield - Improvement or no change in sensory properties up to 40% fat replaced by DE (with gelled W_1_)	[74]
- **Mulberry anthocyanins** (0.5%)	- Walnut oil - PGPR (6%)	- Pectin (1%) - Proclin 300 (0.05%) - GDL (0–2%)	40:60	40:60	- No delamination of DEs after 28 d of storage at 4 °C as compared to 25 °C - Particle size: ↑ from 625 to 1781 and 2316 nm after storage at 4 °C and 25 °C, respectively; - Zeta potential: ↑ from −48 to −40 and −25 mV after storage at 4 °C and 25 °C, respectively; - ↑ Yellowness and ↓ reddishne ssafter storage at 4 °C and 25 °C, respectively; - EE: ↓ from 95.3% to 93.2% and 88.9% after storage at 4 °C and 25 °C, respectively - Rheology: ↑ *G′* and *G″* after the addition of GDL during the frequency sweep test, suggesting gel-like behavior - ↑ 3D printing ability by GDL, particularly at 1.6%	[70]
- WPI (2%) - **Lutein** (0.002%)	- Sunflower oil - PGPR (4%) - **Lutein** (0.002%)	- WPI (2%) - Xanthan gum (0.5%)	10:90	20:80	- Droplet size of DEs: 40–49 µm - ↑ Lutein stability against light - ↑ Lutein bioaccessibility after in vitro digestion in DEs rather than W/O - ↓ Lutein content from 18.8 to 12.3 µg/g and 19.9 to 11.3 µg/g after 14 days for W/O-L/W and W-L/O/W, respectively. - ↑ Lutein recovery (99%) after digestion for W/O-L/W and W-L/O/W - The highest color stability in W-L/O/W	[69]
- Sample A: NaCl (0.1 M), glycerol (3%), ***Opuntia stricta* var. *dillenii*** (OPD extract; 750 mg) - Sample B: **OPD extract** (750 mg), gelatin (6%)	- Sample A: MCT, PGPR (5%) - Sample B: MCT, PGPR (14%), phosphatidylcholine (4%)	- Sample A: NaCl (0.1 M), Tween 20 (2%), - Sample B: NaCl (1.3%), glycerol (13%), and a mixture of caseinate (3%), guar gum (0.175%), and gum Arabic (0.265%)	- Sample A: 25:75 - Sample B 30:70	Sample A: 25:75 - Sample B 30:70	- Higher a* (red to green) value of sample B than sample A Zeta = −32.7 (sample A) and −49.2 (sample B) - The lowest size (1.75 µm) and stability for sample A - EE: 68.2–98%, sample B was more efficient - In vitro gastro-intestinal digestion: ↑ and ↓particle size for sample A and B, respectively. - ↑ Bioaccessibility of the individual betanins and phenolic compounds after encapsulation (67.1 to 253.1%) in comparison with the non-encapsulated ones (30.1 to 64.3%), except for neobetanin	[75]
- Gelatin (10%) - **Phycocyanin** (0.2%)	- Soybean oil - PGPR (4%) - **Astaxanthin** (2%)	- SC (3%) - Gellan gum (0.1%, 0.2%, 0.3%, 0.4%, 0.5%, and 0.7%)	20:80	60:40	- Droplet size: ↓ from 14.98 μm for DE without gellan gum to 7.11 μm at a concentration of 0.7% due to the ↑ viscosity of W_2_ -Rheology: ↑ *G′* and ↑ *G″* after gellan gum addition - ↑ Water holding capacity at higher gellan gum concentrations - ↓ serum separation, ↑ ionic and heat stability at gellan > 0.3% - ↑ EE with ↑ the concentration of gellan gum (90.82% for astaxanthin at 0.5% and 94.1% for phycocyanin at 0.7%) - ↑ Color stability as evidenced by no color change after 10 days In vitro release: <25% for phycocyanin and astaxanthin in SGF and >60% in SIF - Successful pH-controlled release - Significant ↑ bioaccessibility of phycocyanin (12.54%) and astaxanthin (14.27%) in DE and DE gels	[13]
- Gelatin (5%) - **Betanin**	- MCT - PGPR (5%) - **Curcumin** (0.75 mg/mL)	- Sugar beet pectin-bovine serum albumin Pickering nanoparticles (0.5−2%)	20:80	10:90 to 90:10	- D_4,3_: ↓ from 95.7 to 34.8 μm by ↑ Pickering nanoparticles from 0.5% to 2%, respectively. ↑ by increasing volume fraction of primary emulsion; - Rheology: change from liquid-like behavior at 0.5% and 1.0% Pickering nanoparticles to gel-like behavior - EE: 84.1% and 65.3% for curcumin and betanin, respectively; - LE: <20% and >20% for curcumin and betanin, respectively; - ↑ Color stability in DEs as compared to free forms of colorants - ↑ Storage stability in DEs (57.9% to 81.5% (curcumin) and 43.5% to 76.6% (betanin)) as compared to free forms - ↑ Extent and rate of FFA released after encapsulation as compared to free MCT - ↑ Bioaccessibility of betanin (42.7%) and curcumin (53.5%) as compared to free forms	[71]
- **Grape seed proanthocyanidin** (GSP; 2 mg/mL) - Sucrose (3%)	- Olive oil - PGPR (5%)	- WPI (3%) - Konjac glucomannan (KGM) (0–1.75%)	30:70	30:70	- ↑ WHC, rheological and texture properties after KGM addition - ↑ Heat stability of GSP with ↑ KGM concentrations - Freeze–thaw stability: ↓ syneresis and GSP retention with up to 1.5% KGM - The highest UV stability 1.5% KGM - In vitro digestion: ↓ hydrolysis of protein and oil droplets and ↑ bioavailability of GSP after KGM addition - ↑ EE and encapsulation stability and ↓ LE, of GSP after 14 days with ↑ KGM - Color: ↑ *L** and *b** values and ↓ *a** values with ↑ KGM	[76]

The * in L*, a*, and b* is a symbol of color parameters.

### 2.6. Fat Reduction Purposes

Since a fraction of the oil phase is replaced by the dispersed water droplets (W_1_), DEs can be utilized to decrease the fat content of food products. The dispersed phase volume fraction and the particle size distribution of DEs can be modulated in such a way that low-fat DEs exhibit sensory and physicochemical properties similar to full-fat simple emulsions. However, the utilization of DEs for fat replacement and the development of low-calorie products has not been extensively studied. Details of studies regarding the fabrication of low-fat foods using the DE system are presented in Table 7. Zhang, et al. [77] used a gel/oil/water (G/O/W) DE as a fat replacer of pork oil in the formulation of emulsified sausages. Gellan gum was used as the gelling agent of the inner phase. The droplet size of the G/O/W was 5.38 µm, close to that of the fat particles in common sausage (5.5 µm). Comparing sausage samples prepared either from pork oil (20% fat, 20F) or G/O/W DE (12% fat and 20% DE, 20DE), and with excess water (8%, 8W) as fat replacers, the 20DE sample had the lowest fat content and highest water content. The energy values for 20F, 8W, and 20DE were 13.55, 9.48, and 7.91 Kj/g, respectively. While the cooking loss of 20DE was higher than the two other samples, their compression loss was not significantly different. Texture analysis of the samples showed that the springiness and cohesiveness of 8W and 20DE were not significantly different from 20F. Adhesiveness and chewiness in the presence of 8W were equal to those of 20F and lower than 20DE and the hardness of this sample of 8W was lower than 20F and higher than 20DE, concluding that partial fat replacement by G/O/W would not affect the textural properties significantly. In a study by Rakshit and Srivastav [78], DE containing hydrolysable tannin (HT) was used as a fat replacer in short-dough biscuits. Different samples, including short-dough without HT and DE (control), short-dough containing unencapsulated HT, and short-dough samples containing HT-loaded DE replacing 20%, 40%, and 60% of the fat, were formulated and assessed in terms of sensory properties and storage stability using fuzzy logic and quantitative analysis. Adding free HT caused a reduction in the diameter and width of the biscuits after baking due to the interaction between HT and proteins. More than a 24% cooking loss of encapsulated HT was observed in the samples. Based on the results of sensory evaluation, the obtained score of the sample with 40% of fat replaced was higher than the other samples and the lowest score was related to the sample containing free HT due to the astringent flavor of tannin, indicating the ability to use DE as an efficient carrier in masking astringent flavors. The shelf life of the sample containing the 40% fat replacer, as the sample with the highest acceptance, based on measuring moisture absorbance, was 98 days. No changes in color occurred during the storage time; however, hardness and HT decreased. This study indicated the ability of DE to formulate low-fat and fortified biscuits. Zhao, et al. [79] used different concentrations of DE (5%, 10%, 20%, and 30%, termed D5, D10, D20, and D30, respectively) as fat replacers in almond-based yoghurt and compared their effect with their counterpart control samples (without DE, termed C5, C10, C20, and C30) on different properties of the product. The results revealed that by increasing DE from 5% to 30%, the water holding capacity (WHC) increased and D20 and D30 had significantly higher WHC than their control counterparts. A positive correlation was also observed between WHC and gel strength. The addition of a DE (at 20% and 30%) reduced syneresis of the yoghurt samples compared to the controls and increasing DE content resulted in a significant decrease in syneresis from 9.77% for D5 to 2.58% for D30. Higher DE content also led to higher hardness value of the gel. Therefore, the D30 sample showed a significantly harder gel and higher viscosity than C30. Sensory analysis of D30 (as the sample with the most appropriate physical properties) and C30 showed equal overall acceptance for both samples, indicating that despite a considerable decrease in the fat content, the consumers could not distinguish differences between the almond-based yoghurt samples with and without DE.

According to these results, using DE in a variety of food formulations as a fat replacer can decrease calories without changing or even improving (at an appropriate replacement percentage) the sensory properties of the final products, and can also offer opportunities for food fortification by providing blends of unsaturated fatty acids in agreement with dietary recommendations.

### 2.7. Improved Edible Packaging Quality

Today, edible coatings or films are considered ideal alternative eco-friendly primary packaging materials for the replacement of synthetic plastics. The potential applications of edible coatings or films as efficient barriers to gas, microbial and chemical contamination, as well as to improve sensory perception and extended shelf life, have been well documented. Due to the highly hydrophilic nature of biopolymer-based edible films, emulsion-based edible films have gained more attention for their better functionality, particularly in terms of mechanical and moisture barrier properties as well as their capacity to deliver and protect sensitive bioactive material in the film matrix [84,85]. At present, there is a limited number of publications on the preparation of edible films based on DE strategies as compared to SE approaches (Table 8). In this connection, Ghiasi and Golmakani [84] investigated the preparation of a novel design of Persian gum-based films functionalized by crocin and cinnamaldehyde using SE and DE methods. In terms of the visual aspects, the addition of DE droplets had higher effects on opacity than SE, due to the larger droplet size. The effects of DE addition in terms of reducing the moisture content, solubility, swelling, and water vapor permeability of edible films were more pronounced in comparison with SE, which was mainly attributed to the more homogenous distribution of emulsion droplets. This result was in good agreement with the higher enhanced mechanical properties and smoother surface after the addition of DE droplets. In contrast, the addition of bioactives to both the free and SE forms led to a weaker texture and rougher microstructure. Moreover, higher protection effects of DE droplets on the storage stability of encapsulated crocin in the W_1_ as compared to SE confirmed its higher antioxidant activity after 14 days. In addition, DE was a more effective strategy to improve the pH and thermal stability of crocin and cinnamaldehyde as well as to inhibit the light degradation of crocin in the film matrix due to the secondary encapsulation and presence of a thicker interfacial layer. In another study, Tessaro, et al. [86] investigated the effect of the addition of “pitanga” leaf extract-loaded DE on the different properties of films based on gelatin, chitosan, and a gelatin–chitosan composite. Visually, the addition of the DE increased ΔE and opacity for all films, indicating that these films were more intense in color. SEM micrographs revealed some migration of the oil droplets to the surface during drying resulting in more heterogeneous structures. All films showed high UV/Vis light barrier properties, especially for those incorporating DE due to the light dispersion effect of the oil droplets. Incorporating a DE resulted in a high Folin–Ciocalteu reducing capacity and antioxidant activity (based on ABTS^•+^ and FRAP) of the films, suggesting the effective encapsulation and protection of extracts by the DE. However, the low concentration of the encapsulated extract, as well as the limitation of its diffusion from the film matrix into agar, contributed to the non-formation of an inhibition zone during the antimicrobial activity study.

In conclusion, incorporating a DE into edible films and coatings can reinforce their mechanical and barrier properties (against water vapor, light, pH, etc.) and make it possible to fabricate active packaging incorporating sensitive colorants and bioactive components, maintaining their properties for a longer time. However, further studies are required on the application of these new generations of edible films on real food products to investigate their efficacy in terms of extending the foods’ shelf life.

## 3. Food Applications of O_1_/W/O_2_ DEs

During the last decade, fewer studies have been conducted on O_1_/W/O_2_ emulsions in comparison with W_1_/O/W_2_ emulsions and examples of their commercial application in the food industry are still limited (Table 9). However, investigation of O_1_/W/O_2_ DEs is highly desirable due to their potential to replace the traditional W/O emulsions, resulting in lower contents of saturated and trans fats without sacrificing mouthfeel. Moreover, the delivery, protection, and controlled release of sensitive bioactive compounds is possible with O_1_/W/O_2_ DEs [89,90]. In this context, Yang, et al. [91] formulated an astaxanthin-loaded O_1_/W/O_2_ DE stabilized by native corn starch (5% and 7%) and high amylose corn starch at 60% and 75% amylose contents. The highest (97.95%) and the lowest (94.52%) values of EE were reported for the 7% native corn starch-based DE and the high-amylose starch-based DE at 75% amylose contents, respectively. This confirmed the efficient protection of astaxanthin in all samples. Similarly, the 7% native corn starch sample presented the best astaxanthin stability (90%) in O_1_/W/O_2_ DE after 35 days of storage followed by those formulated with a 5% native corn starch (74%) and high-amylose starch-based DE at 60% (69%) and 75% (42%) amylose contents, respectively. This fact pertained to the smaller droplet size of the 7% native corn starch-based DE, which led to a slower release of astaxanthin. According to an in vitro release study, the formation of a strong starch shell resulted in controlled release properties from the DEs (<10% and 50% in gastric and intestinal fluids, respectively) as compared to the free-form solution of astaxanthin with a burst release (closely to 60%) in gastric fluid after 1 h, followed by a slower release in intestinal fluid up to 86%. The controlled release behavior of the O_1_/W/O_2_ DEs was more propounded in the 7% native corn starch-based DE due to the formation of a denser and more rigid interface after starch gelatinization. The diffusion releasing mechanism was also reported as the best kinetic model for astaxanthin release. The obtained results showed that the release rate of astaxanthin depended more upon the droplet size, DE structure, and gel structure of the starch than the hydrophobic and starch–astaxanthin complex interactions.

O_1_/W/O_2_ DEs were also studied for co-encapsulation of selenium-enriched (Se) peptides and vitamin E [1]. As the Se-peptides increased, the emulsion loading of both Se-peptides and vitamin E decreased, mainly due to the disruption of the DE structure. Conversely, the increasing vitamin E content had no significant effect on Se-peptide loading. However, the higher content of vitamin E caused an initial increasing trend of vitamin E loading, accompanied by a gradual decrease. This fact was attributed to the very high viscosity of the inner oil phase at a high concentration of vitamin E followed by increasing the osmotic pressure within the emulsion droplets, resulting in swelling and instability. According to a rheological study, encapsulation did not affect the shear-thinning nature of DEs. However, it decreased the apparent viscosity and the viscoelasticity of the DE, leading to lower stability. Unfortunately, O_1_/W/O_2_ DEs did not affect the odor masking properties of the Se-peptides after encapsulation due to the excessive oil content of the emulsion, which enhanced the unpleasant rancid odor. However, electrical noise results showed the effective potential of O_1_/W/O_2_ DEs for masking Se-peptide odor. In another study, O’Dwyer, et al. [92] developed spreads based on O_1_/W/O_2_ DEs consisting of different omega-3-rich oils (camelina oil and tuna oil) in the inner oil phase. Lipid peroxide values slightly increased during cold storage due to the high polyunsaturated fats, while the p-Anisidine content remained constant. It should be noted that O_1_/W/O_2_ DEs presented lower values of p-Anisidine as compared to non-encapsulated omega-3-rich oils. Both DEs formulated with only camelina oil and the mixture of camelina oil and tuna oil had more oxidative stability than those formulated with tuna oil. Sensory evaluation of the spread reported lower scores in odor and flavor of the formulations containing only tuna oil or high concentrations of tuna oil in the mixture of oils. The highest general acceptability was achieved by the camelina-based O_1_/W/O_2_ DE and the control spreads (W/O emulsion). In another work by O’Dwyer, et al. [93], lipid oxidation, rheology, and firmness were also investigated. They reported that oil with higher unsaturation caused higher p-Anisidine, but did not impact the peroxide level. The blend of fish oil and camelina oil (in a 95:5 ratio) had the highest storage modulus and hardness value. Sensory evaluation of the spreads indicated that the fish oil and camelina oil, at 5:95 and 15:85 ratios, obtained the highest acceptance after storage at 5 °C for 8 weeks.

Katsouli, et al. [94] used O_1_/W/O_2_ DE to encapsulate co-enzyme Q10 and conjugated linoleic acid (CLA). Extra virgin olive oil (EVOO) or olive pomace oil (OPO), and polyphenols from olive kernels were used in this research. Based on the results, loading CLA and polyphenols led to smaller droplets due to their easier homogenization. Comparing the two oil types, EVOO resulted in smaller droplets due to its lower tension. The viscosity of the O_1_/W/O_2_ DE samples was in the range of 74.3–94.3 cP, with the lowest value pertaining to the Q10-loaded DE containing OPO. After storage at 25 °C, the particle size increased more than 500 nm in all samples, indicating coalescence; however, the DE with EVOO showed the highest stability with the smallest size increment. The retention values of CLA and Q10 after 30 days of storage were more than 80%, indicating appropriate stability and encapsulation. Antioxidant activity decreased during storage, which was more pronounced at 25 °C compared to 4 °C. Among the different samples, the highest antioxidant activity belonged to the DE with Q10 and polyphenols. In a study by Lin, et al. [95] alginate-based emulsion microgels were fabricated using the gelation of the inner or outer phases of an O_1_/W/O_2_ DE, and then used to encapsulate lycopene. Based on the results of in vitro digestion, lycopene was released after 1–2, 2–3, and 3.5–4.5 h in SIF from the emulsion gel beads with (6%) and without emulsion microgel particles prepared through gelation of the inner/outer O_1_/W/O_2_ phases. However, in the presence of the microgel particles made through the gelation of the outer phase, slower release occurred due to the denser structure of the gel beads.

Although many studies have been carried out on W_1_/O/W_2_ DEs, there is little such research related to O1/W/O_2_ DEs. O_1_/W/O_2_ DEs are considered advanced design approaches for oil structuring with promising applications in the food industry. The possible innovative food applications of O_1_/W/O_2_ DEs, as a means of replacing conventional W/O emulsions with lower fat content, better sensory properties, higher release control, prolonged retention, and the ability to co-encapsulation both hydrophilic and hydrophobic compounds, have been investigated only in a limited number of studies. Therefore, O_1_/W/O_2_ DEs may become a major research trend within the food industry.

**Table 9 foods-13-00485-t009:** Overview of the past decade’s publications on different food applications of O_1_/W/O_2_ double emulsions (DEs).

O_1_	W	O_2_	Phase Ratio		Remarkable Results	Reference
			O_1_:W	O_1_/W:O_2_		
- Sunflower oil - **Vitamin E** (1–20 mg/mL)	- OSA starch (1–8%) - **Selenium-enriched peptide** (Se-peptide; 5–25 mg/mL)	- PGPR (2–10%) - Sunflower oil	10:90 20:80 30:70 40:60 50:50	30:70 40:60 50:50 60:40 70:30	- ↓ Particle size at 1 and 2% OSA at OSA mass fraction < 4% - ↑ Particle size and zeta potential with ↑ PGPR and ↑ particle size up to 6% OSA and ↓ zeta potential with ↑ OSA - ↓ Loading of both Se-peptide and vitamin E with ↑ Se-peptide - DPPH scavenging rate of 42% and tyrosine inhibition rate of 87.5% for Sep/VE-loaded DE - Not effective in odor masking of Se-peptide - Shear thinning behavior - ↓ Viscosity, viscoelasticity, and stability by loading Se-peptide/V_E_	[1]
- **Curcumin** (1%) - Sunflower oil	- Lecithin (6%)	- Sunflower oil	18:82	85:15	- ↑ Particle size from 392.5 nm to 663.6 after 7 days - ↑ Release from 0.11% to 17% and 0.18% after 14 and 21 days, respectively; - Emulsion stability index: 55.8% after 21 days	[96]
- **Astaxanthin** (0.3%) - Soybean oil - Span 80 (1.5%)	- Tween 20 (1.15%) - Different emulsifiers including native corn starch (NCS; 5–7%) and high amylose corn starch with 60% amylose content (HAS_60%AC_, 5%), and HAS_75%AC_ (1%)	- Soybean oil - Span 80 (5%)	30:70	60:40	- EE range: 94.52–97.95% - The highest storage stability for 7% NCS - ↑ Stability with ↓ size - Particle size: AST-loaded 5% NCS DEs, AST-loaded 7% NCS DEs, AST-loaded HAS60%AC DEs, and AST-loaded HAS75%AC DEs: 10.22, 8.05, 11.22, and 31.81 μm, respectively; - The highest and lowest AST content after 35 days: AST-loaded 7% NCS DEs and AST-loaded HAS75%AC - 86% release for free AST (max) and 27% for 7% NCS DEs	[91]
- Extra virgin olive oil (EVOO) - Olive pomace oil (OPO) - **Conjugated linoleic acid** (CLA) (6%) - **CoQ10** (6%)	- Polyphenols (PP) of olive kernel (1%) - Tween 40 (6%)	- EVOO/olive pomace oil (OPO) - Span 20/Tween 40 (2%)	12:88	3:97	- ↓ Particle size by EVOO and incorporating PP and CLA and ↑ size during storage - Facilitate homogenization after PP addition - Viscosity: 74.3–94.3 cP - The highest stability for sample with EVOO - ↓ Oxidative activity during storage - ↑ Total phenolic content and antioxidant activity during storage of DEs than O/W - The highest oxidative activity for samples with Q10 and PP - Bioactive retention values > 80% after 30 days at 4 °C or 25 °C	[94]
- Camelina oil/ fish oil/blend	- NaCl (6%) - SC (6%)	- Palm oil - Sunflower oil- PGPR (0.5%) - β-carotene (0.016%)	50:50	50:50	- ↑ Particle size by ↑ fish oil content (D_0.5_ = 0.97 at 100% F) - ↓ Oxidation by ↑ Camelina oil content in blended oil - ↑ PV with no changes in p-Anisidine during storage - Masking fishy odor at higher Camelina oil content - Acceptable sensory score for 15:85 and 5:95 camelina:fish blends after 8 weeks - The highest *G′* and hardness at a 5:95 camelina:fish blend ratio	[93]
- **Omega-3** rich oils including cameninal oil, tuna oil, and their blends	- NaCl (6%) - SC (6%)	- Palm oil:sunflower oil (42.65:57.36 ratio) - PGPR (0.5%) - β-carotene (0.016%)	50:50	50:50	- ↓ Oxidative stability by ↑ PUFA - The highest oxidation in the tuna spread - ↑ PV during storage - Similar odor to control at higher tuna oil content in the blend - Higher odor and flavor score for control spread - Higher general acceptance for tuna oil and control spread - No flavor acceptability after 8 weeks for all samples	[92]

## 4. Conclusions

During the last decade, extensive studies have been conducted on the development and novel applications of DE systems, particularly W_1_/O/W_2,_ in the food sector. In this regard, several reviews have focused on basic knowledge of the design, emulsification approaches, stabilization mechanism, and properties of DEs as affected by different parameters, such as the type and concentration of emulsifiers, as well as phase ratio. However, our main focus in this review is to assess specific applications of both W_1_/O/W_2_ and O_1_/W/O_2_ DEs in different aspects of the food industry with emphasis on research published within the last decade. Generally, increasing consumer demand for and consumer of the positive impacts of nutraceuticals on human health is the main driving force behind the development of functional foods. Conventional DEs and their novel designs, using the sol-gel and Pickerng mechanisms, are among the best candidates for the fortification of liquid, solid, and semi-solid foods with both hydrophilic and hydrophobic nutraceuticals. Therefore, the most widespread applications of DE systems in the food sector have been related to the encapsulation of sensitive bioactive compounds to increase the nutritional value of foods and enhance their stability during passing through the digestive tract. However, investigations of DE incorporation into reduced-fat foods, the masking of unpleasant flavor characteristics, and the enhanced color, oxidation stability, and preservation effects of DEs in food formulations, are still limited. Moreover, there are only a few publications on the protection and delivery of enzymes, and probiotics, as well as the incorporation of DEs into edible packaging materials to enhance their physicochemical properties. Nevertheless, there are still some significant challenges that need to be addressed before the successful application of DEs into the food industry. For instance, most preparation methods cannot be readily scaled up. Extending the physicochemical stability of the DEs during storage and also after the exposure of the food or under preparation conditions are also concerns. Optimizing the ratio of hydrophilic/hydrophobic emulsifiers is one of the challenges in terms of stabilizing DEs. Additionally, the requirement for non-food-grade components, particularly hydrophobic emulsifiers, also limits their application for human consumption. DEs are usually prepared through two-step and one-step emulsification processes. Each of these two methods has its limitations, the former is a complicated method, and, in the latter, the encapsulation efficiency is low. Further research is necessary to develop highly stable DEs and to fully investigate the utilization opportunities of incorporating DEs into the complex matrixes of real food formulations and their behavior during long-term storage. Therefore, this review provides new insights into the applications of double emulsions in food matrices and can represent a set of guidelines for the development of healthier food formulations.

## Figures and Tables

**Figure 1 foods-13-00485-f001:**
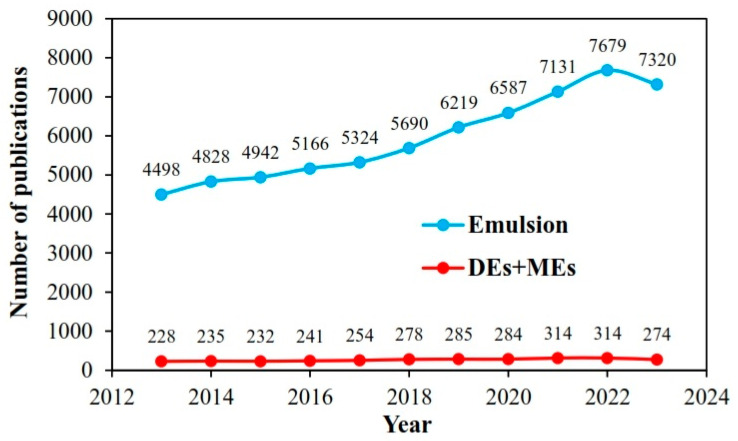
The number of publications on double emulsions (DEs) or multiple emulsions (MEs) among all publications on emulsions over the last decade, identified by the keywords “emulsion”, “DE”, and “ME” (the data were obtained from Scopus database by searching in the title, abstract, and keywords sections).

**Figure 2 foods-13-00485-f002:**
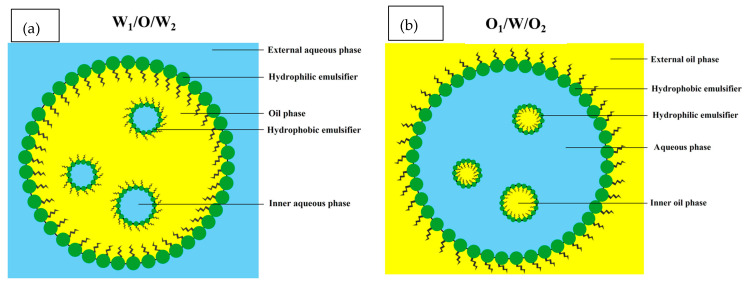
Schematic diagrams of (**a**) W_1_/O/W_2_ and (**b**) O_1_/W/O_2_ double emulsions.

**Table 4 foods-13-00485-t004:** Overview of the past decade’s publications on enzyme encapsulation by W_1_/O/W_2_ double emulsions (DEs).

W_1_	O	W_2_	Phase Ratio		Remarkable Results	Reference
			W_1_:O	W_1_/O:W_2_		
- **Nattokinase** (20%)	- Soybean oil - PGPR (6%)	- SPI (1%) - Polyglutamic acid (PGA) (1%) - Complex SPI:PGA (1%) at different ratios (5:1, 3:1, 1:1, 1:3, 1:5)	30:70	30:70	- Smaller droplet size of complexes than individual SPI and PGA - Higher zeta potential of complexes than individual SPI and PGA by stronger repulsive force - Highest apparent viscosity for the 1:5 ratio complex - Highest EE (97.19) for the 1:3 ratio complex - ↓ Release rate of FFA for in vitro simulated digestion for complex-stabilized DEs - Highest bioavailability (80.69%) for the 1:3 ratio complex	[51]
- **Lactase** (100 U) - Potassium phosphate buffer (0.02 M)	- Corn oil - PGPR 90 (0.5%)	- Gelatin (5%) - Gum Arabic (5%)	33:66	- Not mentioned	- Optimum core solution concentration: 1% - ↑ Storage stability due to the low water activity (≤0.4) and particle size (≤93.52 μm) - ↑ EE (≥98.67%) - ↑ Significant pH stability, temperature stability, and storage stability of enzymes compared to the free-form - ↓ and ↑ of both release rate and lactase activity in SGF and SIF, respectively, after encapsulation	[54]
**Nattokinase** (20 mg/mL)	- Different oils (MCT, liquid paraffin) and emulsifiers (10–25%) including Abil EM90, Arlacel P135, andSpan 80	Labrasol (10–20%)	30:70 40:60 70:30	10:90 20:80 30:70 40:60 50:50 60:40 70:30 80:20 90:10	- ↑ Emulsifying capacity of MCT as oil phase and EM90 as emulsifier - Optimized condition: 20% Abil EM90, 15% Labrasol, and 40% W_1_ - ↓ D_4,3_: 5.3 to 4.7 μm after 30 days - Initial EE: 86.8% - ↓ EE to 82.6% after 30 days at 25 °C - Improved sustained in vitro release (30% after 8 h) - The release rate in vitro under pH values: 1.21 > H_2_O > 4.50 > 6.80 - ↑ Blood clotting time in mice at all doses (111.3–194.1 s) as compared with free forms - Enhanced carrageenan-induced tail thrombosis outcome	[53]

**Table 5 foods-13-00485-t005:** Overview of the past decade’s publications on probiotics encapsulation by W_1_/O/W_2_ double emulsions (DEs).

W_1_	O	W_2_	Phase Ratio		Remarkable Results	Reference
			W_1_:O	W_1_/O:W_2_		
- ***L. acidophilus*** (5.1 × 10^7^ CFU/g) - NaCl (0.15 M)	- MCT oil - Fish oil (0–10%) - PGPR (1–5%)	- Different emulsifiers including SPI, (0.5–2%) and SA (0.25–1%)	10:90 20:80 30:70 40:60	30:70 40:60 50:50 60:40	- ↑ W_1_ from 10% to 30% led to ↑ stability—the only stable DE in 7 days at 4 °C: at 10% W_1_ - ↑ Fish oil and SA led to ↑ viability in SGF and SIF - ↓ GI release by SA and fish oil - ↑ Cell count and ability to adhere to the intestinal mucosa by the addition of fish oil - ↑ Viscosity, stability, and probiotic EE in the presence of SA - The highest EE 0–7.5% fish oil, 1.5% SPI, 0.27–0.75% SA - ↑ SPI and ↑ SA led to ↑ viscosity - ↑ SA led to ↓ size	[58]
- ***L. plantarum*** (11 log CFU/g) - Different emulsifiers including gelatin (2%), - Alginate (2%)/CaCl_2_ (100 mM), tragacanth gum (2%), carrageenan (2%)/KCl (100 mM)	- Olive oil - PGPR (6%)	- Tween 80 (4%)	20:80	40:60	- Size range: from 6.4 (tragacanth) to 14.7 µm (alginate) - Zeta range: −21.1 to −46.2 mV - The least stability was for carrageenan (>80%) - The highest EE (97.4%) by carrageenan - ↑ Heat protection by gelling agents - Best gelling agent: tragacanth due to increased viability at low pH and heating to 28.05% and 16.74%, respectively - The highest viscosity by alginate	[60]
- ***L. plantarum* F1** - Different prebiotics including mannitol (2%) and trehalose (2%) - Guar gum (0.5%) - Tween 80 (5%)	- Sunflower oil - Lecithin (5%) - Lipophilic sea buckthorn pomace extract (LSBPE, 5%)	- Alginate (2%) - Tween 80 (5%)	40:60	40:60	- Best condition: Mannitol as prebiotic due to high encapsulation yield (82.19%), good cell survival rate (76.99%), and low chemical degradation of the oil (PV: 3.8 meq O_2_/kg fat) after 42 days Size: 2.2–2.3 µm - ↑ Oxidative stability by LSBPE - ↓ 1.02 and 5.79 log encapsulated and free cells in SGF, respectively - ↑ Gelation time led to ↑ cell count and gel hardness	[59]
- ***L. plantarum*** (10.4 log CFU/g) - Sucrose (10 g/L) - Glucose (4.6 g/L)	- Corn oil - PGPR (1.5%)	- Tween 80 (0.75%) - Tara gum (1.5%)	30:70	30:70	- Size: 12 µm - Pseudoplastic behavior (index 0.63) - EE: 86% (7.92 log CFU/mL) - Results of mango dessert incorporating 25% DE: - ↑ Cell by glucose and lactose - ↑ Viability in DE - Low protection in the small intestine - 3.85 log CFU/mL count in large intestinal	[61]
- ***L. reuteri*** (10.69 log CFU/mL)	- MCT - PGPR (5%)	- Poloxamer 407 (2.5%)	20:80	20:80	- Size: 13.4 µm, no changes in size in 30 days - EE: 7.23 Log CFU/mL during cold storage - ↓ Stability lower than control (from 6.18 to <1 Log CFU/mL) compared to encapsulated bacteria (from 7.23 to 2.82 Log CFU/mL). - 70% survival in GI for encapsulated bacteria - No changes in cell count after 3 days at 6 °C - 5.2 and 2.82 log CFU/mL on the 15th and 30th days	[55]
- ***L. Plantarum*** - Fructooligosaccharides (2%)	- MCT - PGPR (0.5–5%)	- Na_2_EDTA - CaCl_2_ - Alginate (2%) - WPI - EGCG conjugates (0.5 to 5%)	20:80 40:60 60:40 80:20	50:50	- ↓ Size with ↑ PGPR, ↓ oil phase, and ↑particle conjugate - ↓ pH led to ↑ *G′* - Hydrogel state at pH ≤ 4 - Low loss in GI (from 7.79 × 10^7^ to 7.39 × 10^7^ CFU/mL) at PGPR content of 5%	[57]
***Bifidobacterium lactis* subsp. lactis *BB-12*** (11.35 log CFU/g) - Glycose (5%) - Inulin (2%)	- Olive pomace oil - Span 20 (2.5%) - Tween 40 (2.5%)	- Different encapsulating agents including SA (2%), pectin (1%), gelatin (1%), casein (1%), and gum Arabic (1%)	20:80	15:85	- Zeta: (−0.14)–(−3.36) mV - SA as the best encapsulating agent - Results of coating of beads with chitosan with two methods: EE: 72.48–81.68%, 68.6–86.1% survival rate in GI- >10^6^ CFU/g count after 1 month, combining the extrusion/DE emulsification as compared to cells encapsulated through conventional extrusion (survival rate 46.8%) after 15 days - 80% viability at acidic pHs - Higher viscosity (283.4 cP) for alginate-pectin DE	[56]
- ***L. plantarum*** - Aguamiel or sweet whey - Panodan SKD (1.6%)	- Canola oil - PGPR 90 (6.4%)	- Mesquite gum (13.2%) - Maltodextrin DE10 (3.4%) - Gum Arabic (3.4%)	30:70	30:70	- Larger size with aguamiel (than sweet whey) - Small size increment after 14 days - Results of cheese preparation incorporating DE: ↑ Cell viability with DE (compared to free cells in cheese), ↑ heat protection by aguamiel DE (than sweet whey DE) - After melting, ↓ Log CFU/g by 2.18, 1.42, and 1.94 for control cheese, cheese formulated with DE/sweet whey and cheese formulated with DE/aguamiel, respectively, - ↑ Viability at low and high pH values in DE (especially aguamiel DE)	[62]

**Table 7 foods-13-00485-t007:** Overview of the past decade’s publications on fat reduction purposes by encapsulation in W_1_/O/W_2_ double emulsions (DEs).

W_1_	O	W_2_	Phase ratio		Remarkable Results	Reference
			W_1_:O	W_1_/O:W_2_		
*- Monascus* pigment - Flaxseed gum (0.75%)	- Soybean oil - PGPR (6%)	- Pea protein isolate (5%)	20:80	40:60	- ↑ Flaxseed gum led to ↑ size, ↓ instability index and ↓ mean square displacement - Results of sausage properties incorporating 0–30% DEs: Texture: ↓ hardness and ↑ cohesiveness by fat replacing with DE, the highest chewiness and gumminess at a 30% DE level - ↓ Lipid from 11.22% to 5.09%, ↑ protein from 15.77% to 17.02%, ↑ PUFA from 23.36% to 59.63%, ↑ WHC and oxidative stabilitycompared control - ↑ Lightness by fat replacing with DE	[80]
- Full-fat almond emulsion containing almond protein isolate (3.5%), almond oil (4%), and sugar (4.5%)	- Almond oil - PGPR (2%)	- Full-fat almond emulsion containing almond protein isolate (3.5%), almond oil (4%), and sugar (4.5%)	40:60	20:80	- Results of set-type yoghurt-like almond-based gels incorporating 0–30% DE: No differences in size and cohesiveness at different DE contents - ↓ Water holding capacity, ↓ hardness, and ↑ syneresis by ↑ DE content - The highest viscosity at 30% DE - No difference in sensory properties of low-fat (containing DE) and full-fat yoghurt	[79]
- Gellan gum (0.4%) - CaCl_2_ (0.5%)	- Refined pork oil - PGPR (3%)	- SC (0.1%)	40:60	80:20	- Size: 5.38 µm - Results of sausage properties incorporating 0 and 20% DE - ↓ Fat and energy values and ↑ water compared to high- and low-animal fat sausages - Cooking loss (9.63%) less than high-fat sausages - Texture: equal adhesiveness and springiness in all sausages, ↓ hardness than high-fat sausages	[77]
- Hydrolysable tannin (10%) - Phosphate buffer	- Sunflower oil (4%) - Lecithin (2%) - span 80 (2%)	- Gum Arabic (3%)	20:80	40:60	- Results of fat-reduced short-dough biscuits incorporating 0–60% DE: - ↑ hardness, ↓ biscuit height, ↑ spread ratio, ↑ antioxidant capacity, and ↓ loss of hydrolysable tannin with ↑ DE - The highest astringent flavor masking and highest acceptability by replacing 40% fat with DE	[78]
- WPI(10–25%) - Rice protein (RP, 10–25%) - Pumpkin seed protein (PSP, 10–25%)	- PGPR (1%) - Span 80 (1%)	- Milk	40:60	5:95	- Higher viscosity (1.8 Pas) and serum index (5.2%) for RP-stabilized DE - The largest and the smallest size related to RP and WPI-stabilized DE - Results of fat-reduced cheese incorporating 5% DE: ↑ Hardness, ↑ cheese diameter, ↑ oil loss (except in WPI cheese) than full-fat cheese	[81]
- NaCl (0.6%)	- Olive oil - PGPR (6.4%)	- SC (10%) - NaCl (0.6%)	50:50	70:30	- Results of model meet emulsion incorporating 0–30% DE: ↑ Fat replacing by DE led to ↓ jelly and fat separation, ↑ WHC, ↓ total expressible fat - ↑ TBAR after 60 days of storage, the lowest TBAR at 10% fat replacement by DE - ↑ DE led to ↓ hardness, ↑ cohesiveness, and ↓ gumminess and chewiness than full-fat control	[82]
- NaCl (0.6%)	- Olive oil - PGPR (6%)	- SC (0.5%)/ - NaCl (0.6%) - WPC (6%)	20:80	40:60	- ↑ Size by WPC (compared to SC), wider size distribution during storage for SC DE - ↑ Thermal stability and ↑ creaming index at 0 °C compared to 7 °C - Results of meat system incorporating 0–34% DE: ↓ Total fluid released by adding DE - Similar hardness, cohesiveness, and springiness for meat systems with/without DE - ↑ Chewiness in fat reduced WPI DE - ↑ Lightness by adding DE	[83]

**Table 8 foods-13-00485-t008:** Overview of the past decade’s publications on improved edible film properties by incorporating W_1_/O/W_2_ double emulsions (DEs).

W_1_	O	W_2_	Phase Ratio		Remarkable Results	Reference
			W_1_:O	W_1_/O:W_2_		
- NaCl (5.84 mg/mL) - **Crocin** (65 mg/mL)	- **Cinnamaldehyde** (33%) -PGPR (8%)	-WPI (8.5%)	50:50	30:70	- Persian gum-based film: - ↓ Opacity by incorporating DE than free bioactive - ↓ Moisture content, water solubility, WVP, and swelling compared to the free and single-emulsion addition strategy - ↑ Contact angle more than control but less than the free and single-emulsion addition - ↑ Tensile strength and elongation at break as compared to control film and free and single emulsion addition - ↑ UV and visible barrier properties, ↑ photostability of crocin against fluorescent and UV lights, ↑ thermal and pH stability of crocin and cinnamaldehyde, ↑ antioxidant activity after 14 days compared to free bioactives	[84]
- **“Pitanga” leaf hydroethanolic extract** (10%)	- Soybean oil - PGPR (3%)	- Tween 80 (3%) - SC (0.5%)	20:80	40:60	- Gelatin, chitosan, and gelatin–chitosan composite films: - ↑ Opacity - ↓ Roughness, contact angle, solubility, and WVP - ↑ Tensile strength and elastic modulus for gelatin and chitosan-based films but ↓ for gelatin–chitosan film, ↑ elongation at break for both gelatin and gelatin–chitosan films - Inhibition of bacterial growth just below the disks - ↑ Folin–Ciocalteu reducing capacity and antioxidant activity	[86]
- **Pitanga leaf hydroethanolic** (10%)	- Soybean oil - PGPR (3%)	- Tween 80 (3%) - SC (0.5%)	20:80	40:60	- Gelatin, chitosan, and gelatin–chitosan composite films with nanocellulose (NC): ↓ Moisture content, solubility, WVP and ↑ opacity, tensile strength, and elongation at break after DE addition - ↑ FCRC, ABTS•+, FRAP after DE addition - Antimicrobial activity only in G-based film incorporating DE against *S. aureus* in the region of contact of the film - Light barrier order: gelatin-NC/DE > gelatin–chitosan-NC/DE > C-NC/DE > gelatin-NC > chitosan-NC > gelatin–chitosan-NC	[87]
-	- Sunflower oil - Lecithin (0.7%)	- Coffee Byproducts (pectin and cellulose (3:7); 0.8–2.4%) - SC (0.5)	15:85	5:95	- Coffee byproduct-based film: Droplet size: 0.38–1.23 μm under the different times of homogenization - ↑ Thickness (0.15 to 0.25 mm), transparency (3.10–5.28%), WVP (3.76–15.96 g mm/m^2^hKpa), tensile strength (1.26–1.79 MPa), and elongation (3.40–5.20%) with ↑ polymer concentration - Antioxidant activity: (EC50; kg film/mol DPPH): ↑ from 2.47 to 4.35 with ↑ polymer concentration	[88] (2023)

## Data Availability

The raw data supporting the conclusions of this article will be made available by the authors on request.

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
