# Peer review of "An Updated Comprehensive Overview of Different Food Applications of W1/O/W2 and O1/W/O2 Double Emulsions"

_foods, 2024, doi:10.3390/foods13030485_

Round 1

Reviewer 1 Report

Comments and Suggestions for Authors

The article represents a major review of different double emulsion applications and research. The submitted review article cited and discussed the most relevant references in the past 10 years. The content of the paper is suitable for Special Issue: Advances in Food Polymers and Colloids: Processing and Applications. Overall, I consider this to be an interesting paper.

However, the corresponding structure is not adequate for scientific publication within scientific journal. The structure of the article, particularly the presentation of numerous studies and their findings in a tabular format, might overwhelm readers. I believe that more than half of the manuscript consists of tables. Simplification or categorization of studies based on compounds, methods, or outcomes could improve readability and understanding. In addition, consider providing tables as supplementary material.

Consider revising your manuscript towards shorter period of time covered by the corresponding review. Consider 5-year period, taking into account that around 2/3 of cited articles are withing 5-year period, in order to shorten the submitted review article. Furthermore, "Food applications of O1/W/O2 DEs" part could be excluded from the manuscript since it does not represent the core nor specific interest of submitted review article. In order to provide as concise review as possible it should cover only specific topic of interest.

While individual studies' findings are well-detailed, a comparative analysis across different studies is missing. Actually, adding a summarized conclusion at the end of each section, highlighting the main findings and their implications, would provide a clearer takeaway for readers. This could enhance the section's depth and provide clearer insights.

In Table 10 it seems to be a typing error regarding Phase ratio since it should refer to O/W/O and not W/O/W.

Restructuring the conclusion to ensure a more cohesive flow of ideas, possibly by categorizing challenges, research gaps, and proposed future directions, could enhance readability and coherence.

Author Response

Reviewer #1

-The article represents a major review of different double emulsion applications and research. The submitted review article cited and discussed the most relevant references in the past 10 years. The content of the paper is suitable for Special Issue: Advances in Food Polymers and Colloids: Processing and Applications. Overall, I consider this to be an interesting paper.

Response: We appreciate your positive outlook and constructive comments.

-However, the corresponding structure is not adequate for scientific publication within scientific journal. The structure of the article, particularly the presentation of numerous studies and their findings in a tabular format, might overwhelm readers. I believe that more than half of the manuscript consists of tables. Simplification or categorization of studies based on compounds, methods, or outcomes could improve readability and understanding. In addition, consider providing tables as supplementary material.

Response: Thank you for your recommendation. The main goal of the authors in this review has been to focus on the various applications of DEs in the food industry, which have been discussed in the text of the manuscript as much as possible considering the existing limitations. To this end, the categorization of information was based on their applications. To help the readers in this field of area, additional information about the composition of the phases and the results of various physicochemical tests have been added in the form of tables. Therefore, the preparation methods of  DEs and the detailed effects of each compound were not the aims of this study to apply for categorization.

-Consider revising your manuscript towards shorter period of time covered by the corresponding review. Consider 5-year period, taking into account that around 2/3 of cited articles are withing 5-year period, in order to shorten the submitted review article. Furthermore, "Food applications of O1/W/O2 DEs" part could be excluded from the manuscript since it does not represent the core nor specific interest of submitted review article. In order to provide as concise review as possible it should cover only specific topic of interest.

Response: If the review is reduced to the last 5 years, some of the food applications of DEs will be removed or limited to just a few publications. The presentation of the results in the last decade was to offer a frequency distribution of published works on different applications of DEs. Therefore, those applications that have received less attention will be clear for further investigation in the future. Although O1/W/O2 DEs were our interest in this review article, their food applications in the last decade were limited to only those mentioned in Section 3. Therefore, this section was also presented to clearly show their lower frequency distribution as compared to O1/W/O2 DEs.

-While individual studies' findings are well-detailed, a comparative analysis across different studies is missing. Actually, adding a summarized conclusion at the end of each section, highlighting the main findings and their implications, would provide a clearer takeaway for readers. This could enhance the section's depth and provide clearer insights.

Response: As you correctly mentioned, a brief conclusion was added at the end of each section and highlighted in yellow color in the revised manuscript.

-In Table 10 it seems to be a typing error regarding Phase ratio since it should refer to O/W/O and not W/O/W.

Response: Thank you for this hint. This table (now as Table 9) was modified in the revised manuscript. Please check Page 43.

-Restructuring the conclusion to ensure a more cohesive flow of ideas, possibly by categorizing challenges, research gaps, and proposed future directions, could enhance readability and coherence.

Response: The conclusion was modified in the revised manuscript to include challenges, research gaps, and proposed future directions. Please check Page 46.

Reviewer 2 Report

Comments and Suggestions for Authors

The review about double emulsion has been well-organized in this manuscript.

The author introduced the function,  application, comparasion for w/o/w and o/w/o system. It provide a overview for current situation and potential application. However, some points should be paid attention. The details as below,

1. Line 34, the authors mentioned DE and ME, what is the term or defination for ME, what is the difference with DE?

2. In abstract, the authors describe the purpose of DE contain reducing salt and sugar, why didn't mention in the main body?

3. Line 153, the size of word should be uniform.

4. The table can be displayed with 3-line table, and all the tables can be more beautiful. The Remarkable results can be reduced. Additionally, the author can give a conclusive structure and organization with a graph.

5. Does the color belong to sensory attributes?

Author Response

Reviewer # 2

- The review about double emulsion has been well-organized in this manuscript. The author introduced the function, application, comparasion for w/o/w and o/w/o system. It provide a overview for current situation and potential application. However, some points should be paid attention. The details as below.

Response: We appreciate your positive outlook and constructive comments.

-Line 34, the authors mentioned DE and ME, what is the term or defination for ME, what is the difference with DE?

Response: Multiple emulsions are complex systems, termed 'emulsions of emulsions'. Double emulsions are a major type of multiple emulsions, where the outer phase contains both oil and water in the dispersed phase which is either oil or water. The two major types are W/O/W and O/W/O double emulsions. However, with advanced technologies, researchers have revealed different types of double emulsions such as W/W/O, W/O/O, O/W/W, and O/O/W. This statement was added in the revised manuscript. Please check Page 1.

-In abstract, the authors describe the purpose of DE contain reducing salt and sugar, why didn't mention in the main body?

Response: Salt and sugar reduction was discussed in section 2.6. “Improved sensory attributes” and Table 6 in the revised manuscript.

-Line 153, the size of word should be uniform.

Response: Thank you for this point. The font size was checked and edited throughout the text.

-The table can be displayed with 3-line table, and all the tables can be more beautiful. The Remarkable results can be reduced. Additionally, the author can give a conclusive structure and organization with a graph.

Response: The design of all tables was modified based on your comment. Moreover, the results of some publications in Tables were reduced or summarized as much as possible. In fact, the tables were prepared to provide additional information on the composition of the phases, and the results of various physicochemical tests to help researchers in this field of area. Unfortunately, due to the variety of studies and subsequently their different results, it was not possible to display them in the form of a graph.

-Does the color belong to sensory attributes?

Response: As you correctly mentioned, the relevant section on “Improved color stability” was merged with the section on “Improved sensory attributes”. Please check the section 2.5 (Page 26)

Round 2

Reviewer 1 Report

Comments and Suggestions for Authors

The authors corrected the manuscript according to my suggestions. 

Author Response

Thank you for your comment. 

Reviewer 2 Report

Comments and Suggestions for Authors

The authors have made the corrections and responses based on the comments. In the current version, lots of tables with complicated information were listed. I think the authors can improve the quality and make them better. 

Author Response

The authors have made the corrections and responses based on the comments. In the current version, lots of tables with complicated information were listed. I think the authors can improve the quality and make them better.

Response: Thank you for your comment. The information was summarized as much as possible in all tables based on your comments.